# Global patterns of vascular plant alpha diversity

**Francesco Maria Sabatini** [1,2,3] ✉, **Borja Jiménez-Alfaro** [2,4], **Ute Jandt** [1,2], **Milan Chytrý** [5], **Richard Field** [6], **Michael Kessler**[7], **Jonathan Lenoir** [8], **Franziska Schrodt** [6], **Susan K. Wiser** [9], **Mohammed A. S. Arfin Khan** [10], **Fabio Attorre** [11], **Luis Cayuela** [12], **Michele De Sanctis** [11], **Jürgen Dengler** [13,14], **Sylvia Haider** [1,2], **Mohamed Z. Hatim** [15,16], **Adrian Indreica**[17], **Florian Jansen** [18], **Aníbal Pauchard** [19,20], **Robert K. Peet** [21], **Petr Petřík** [22,23], **Valério D. Pillar** [24], **Brody Sandel**[25], **Marco Schmidt** [26,27], **Zhiyao Tang** [28], **Peter van Bodegom** [29], **Kiril Vassilev**[30], **Cyrille Violle**[31], **Esteban Alvarez-Davila**[32], **Priya Davidar** [33], **Jiri Dolezal** [22,34], **Bruno Hérault** [35,36,37], **Antonio Galán-de-Mera** [38], **Jorge Jiménez** [39], **Stephan Kambach** [2], **Sebastian Kepfer-Rojas** [40], **Holger Kreft** [41,42], **Felipe Lezama**[43], **Reynaldo Linares-Palomino** [44], **Abel Monteagudo Mendoza**[45,46], **Justin K. N'Dja**[47], **Oliver L. Phillips** [48], **Gonzalo Rivas-Torres**[49], **Petr Sklenář**[50], **Karina Speziale** [51], **Ben J. Strohbach** [52], **Rodolfo Vásquez Martínez**[46], **Hua-Feng Wang**[53], **Karsten Wesche** [1,54,55] & **Helge Bruelheide** [1,2]

Global patterns of regional (gamma) plant diversity are relatively well known, but whether these patterns hold for local communities, and the dependence on spatial grain, remain controversial. Using data on 170,272 georeferenced local plant assemblages, we created global maps of alpha diversity (local species richness) for vascular plants at three different spatial grains, for forests and non-forests. We show that alpha diversity is consistently high across grains in some regions (for example, Andean-Amazonian foothills), but regional 'scaling anomalies' (deviations from the positive correlation) exist elsewhere, particularly in Eurasian temperate forests with disproportionally higher fine-grained richness and many African tropical forests with disproportionally higher coarse-grained richness. The influence of different climatic, topographic and biogeographical variables on alpha diversity also varies across grains. Our multi-grain maps return a nuanced understanding of vascular plant biodiversity patterns that complements classic maps of biodiversity hotspots and will improve predictions of global change effects on biodiversity.

Our understanding of the global patterns of plant diversity largely stems from studies based on either local to national floras or stacked distribution range maps[1–4]. These approaches allow quantification of the total number of species occurring in a region but do not address how plant species co-occur locally and form species-rich or species-poor communities. With the notable exceptions of trees and ferns[5–9], the global distribution of local plant diversity remains poorly understood[10].

The species richness of local plant communities, i.e., alpha diversity, is non-linearly related to the size of the sampling unit, i.e., the

spatial grain[9,11–13]. Enlarging the sampling unit means that more species are progressively captured in the same plot, so that the alpha diversity of a sampled plot slowly, but non-linearly, approaches the regional species richness, i.e. gamma diversity[11,12,14]. The steepness of the curve, i.e., beta diversity, determines how the plant community composition varies from place to place[11]. This non-linearity complicates direct comparisons of biodiversity data from place to place and makes mapping alpha diversity across large areas challenging. Even in well-sampled regions, available data are heterogeneous mixtures of surveys with varying spatial grains and sampling protocols, and different reference taxonomies[9,15,16]. Furthermore, there is a typical trade-off between spatial grain and extent in biodiversity research, with most fine-grained studies only covering limited spatial extents. Thus the question of whether global patterns of alpha diversity are consistent with known patterns of regional gamma diversity has remained unanswered.

Plant diversity patterns result from ecological and evolutionary processes acting at different spatial and temporal scales[17,18]. At continental and regional scales, evolutionary processes (migration, speciation, extinction) as well as geological and climatic history play key roles[19,20]. At local scales, diversity depends primarily on assembly processes related to species dispersal, habitat filtering and biotic interactions (including humans)[3,21,22]. There is clearly an intimate nested relationship between processes at different scales, as a species must be present regionally to occur locally, and large-scale environmental factors influence local conditions[8,17,23]. An exploration of alpha diversity patterns at multiple grain sizes can discriminate between areas where species richness is consistently high or low across grain sizes, and those where it is not, i.e., where species richness is either high at fine grains and low at coarse grains, or vice versa[9,10,24,25]. This may provide insights into the prevailing mechanisms that shape biodiversity distribution at different scales, and which produce and maintain global plant diversity[11,26]. For example, the discrepancies between alpha diversity patterns at different grains could indicate regional or biome-related variation in the roles of habitat heterogeneity, dispersal barriers or environmental filtering[27–29].

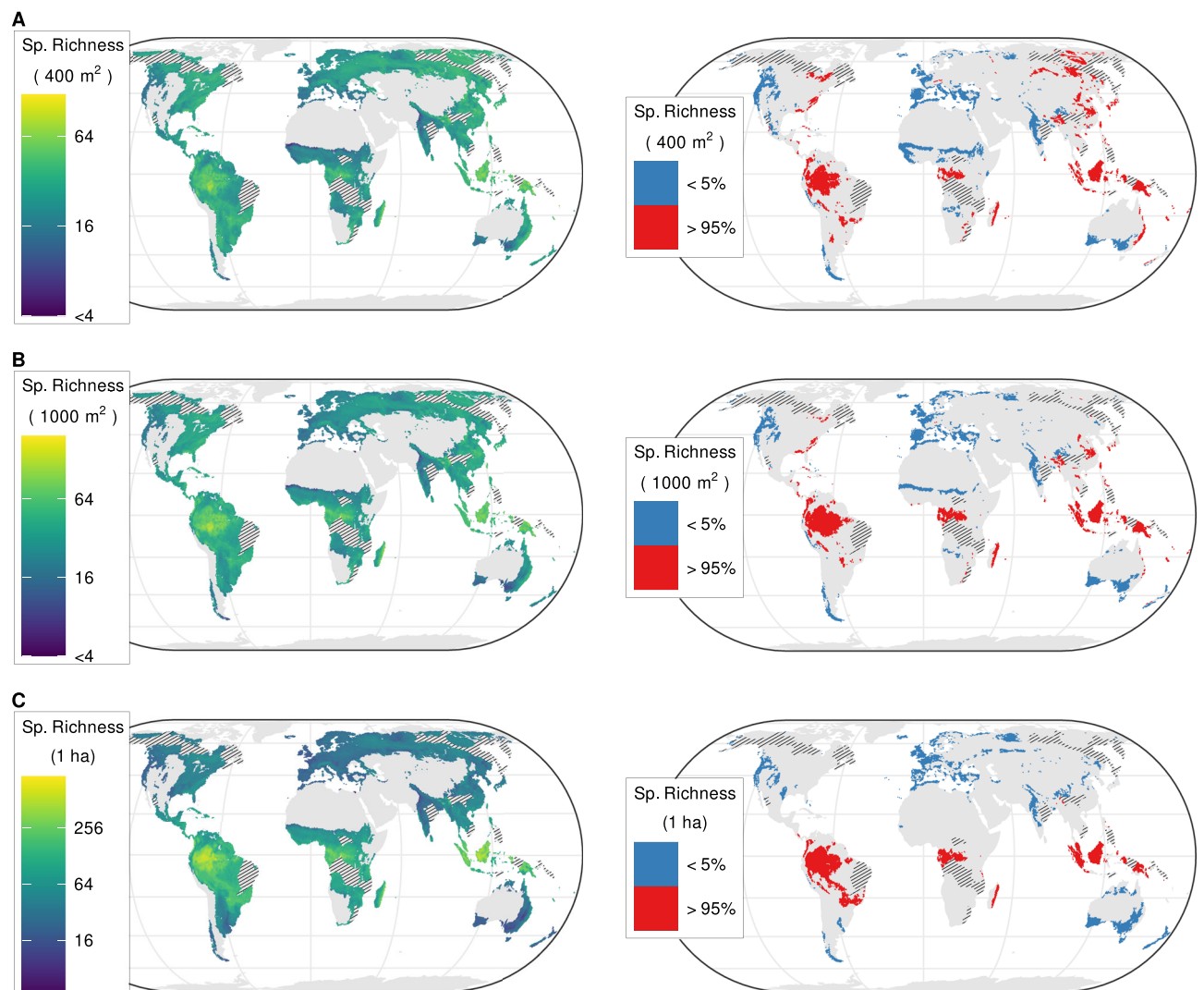

**Fig. 1 | Global distribution of estimated vascular plant alpha diversity in forests.** Spatial grains: **A** 400 m²; **B** 1000 m²; **C** 1 ha. The maps on the left show the median estimated species richness at the corresponding spatial grain for each 2.5 arcminute grid cell of the World, averaged over 99 boosted regression tree models based on different resampled datasets. Colors are on a log₂ scale. The maps on the right show the distribution of hotspots (red) and coldspots (blue), i.e., areas where species richness is above the 95th or below the 5th global percentile, respectively.

We only show alpha diversity estimates for locations where forests would grow under current climate conditions and without human influence[208]. Hatching represents data-poor regions, i.e., regions farther than 500 km from any vegetation plots, for which we did not generate predictions. Global maps with predictions for these data-poor regions can be found in Supplementary Fig. 3. Values are averaged over 2600 km² hexagons. Source data are provided as a Source Data file.

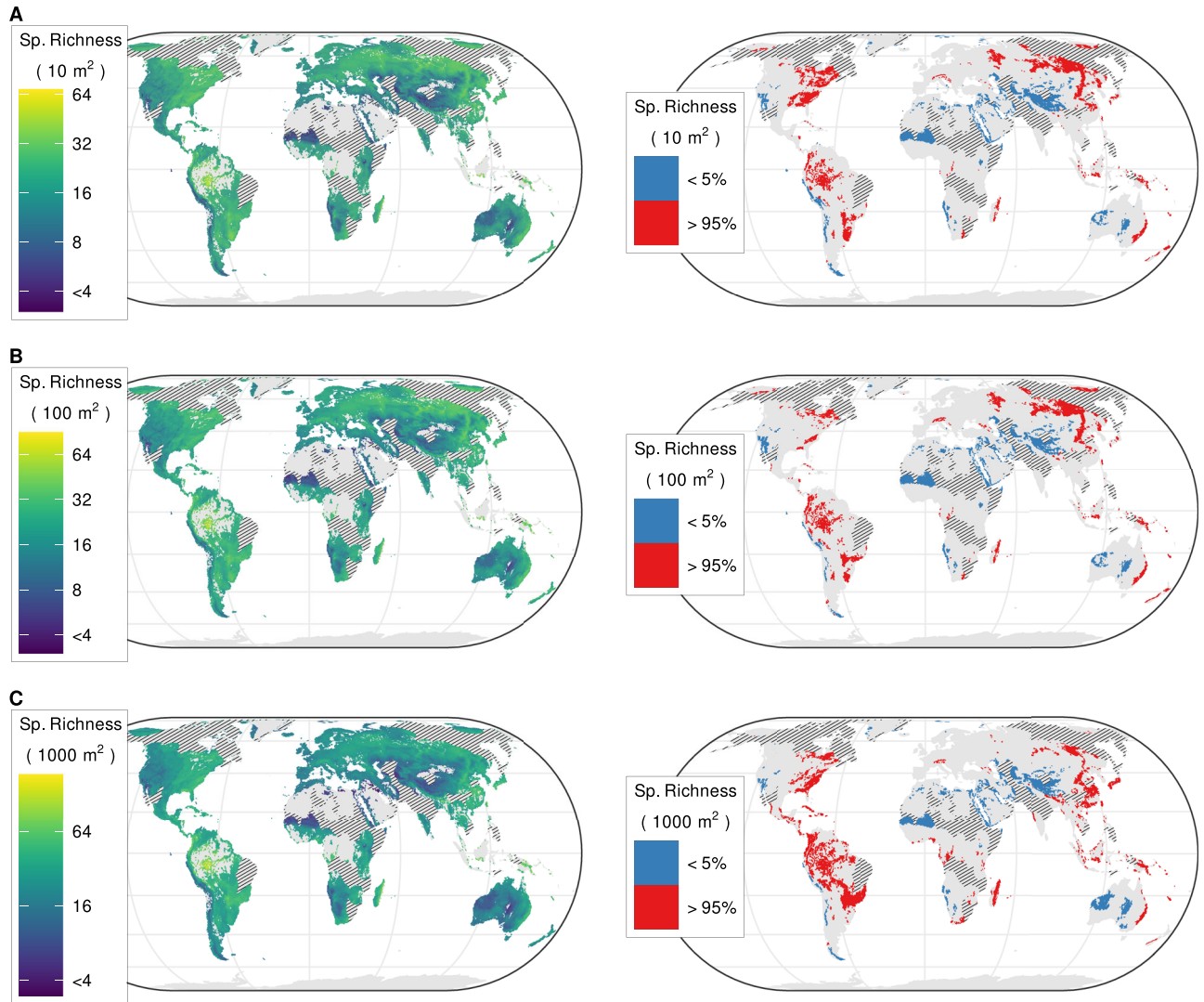

**Fig. 2 | Global distribution of estimated vascular plant alpha diversity in non-forest ecosystems.** Spatial grains: **A** 10 m²; **B** 100 m²; **C** 1000 m². The maps on the left show the median estimated species richness at the corresponding spatial grain for each 2.5 arcminute grid cell of the World, averaged across 99 boosted regression tree models based on different resampled datasets. Colors are on a log₂ scale. The maps on the right show the distribution of hotspots (red) and coldspots (blue), i.e., areas where species richness is above the 95th or below the 5th global percentile, respectively. We only show alpha diversity estimates for locations where the land cover 'herbaceous vegetation' occurs based on a consensus map that integrates multiple global remote sensing-derived land-cover products[209]. Hatching represents data-poor regions, i.e., regions farther than 500 km from any vegetation plots, for which we did not generate predictions. Global maps with predictions for these data-poor regions can be found in Supplementary Fig. 5. Values are averaged over 2600 km² hexagons. Source data are provided as a Source Data file.

Here, we explore alpha diversity patterns across multiple spatial grains globally. We leverage methodological advances in modeling biodiversity across scales[9,12,18,30–32] using the sPlot database, a global initiative that aggregates and harmonizes local-scale species co-occurrence data from hundreds of independent datasets and vegetation surveys[15,16]. The sPlot database incorporates more than 1 million vegetation plots and covers both natural and semi-natural ecosystems on all continents and in all biomes[15]. We focused on terrestrial vascular plants only, since data on bryophytes, lichens, vascular epiphytes and aquatic habitats are too scattered in the sPlot database.

We applied machine learning (boosted regression trees) to model the relationships between vascular plant species richness at different grains and 20 global datasets on current and past climate, soil and topography. Our models allowed relationships between alpha diversity and environmental variables to vary across grains by including interaction terms between plot size and other predictors[9]. To simultaneously quantify uncertainty and to account for the uneven distribution of data across biomes and vegetation formations in our database, we averaged our results over 99 model runs, each based on a stratified resampling of the data (Supplementary Fig. 1, Supplementary Data 1). By modeling the relationships between alpha diversity and environmental variables across the globe, we (1) predicted alpha diversity of vascular plants at three different grain sizes spanning two orders of magnitude, (2) determined how the explanatory power of potential environmental drivers on alpha diversity varies across the three grain sizes, and (3) identified regional scaling anomalies, i.e., areas where alpha diversity is high at fine grain but low at coarse grains, or vice versa.

## Results
### Multi-grain global maps of local species richness
We modeled forest and non-forest ecosystems jointly but focus on each broad formation separately in the main text. Modeling them separately yielded similar results (not shown). For forests, we generated estimates for the three grain sizes most commonly used for sampling forests: 400 m², 1000 m² and 1 ha. At the finest grain

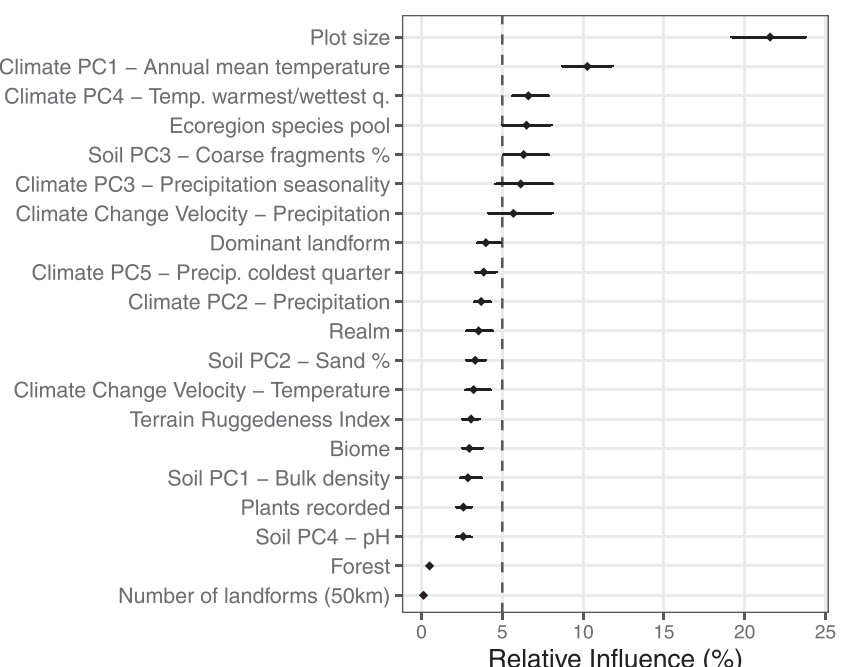

**Fig. 3 | Relative influence of environmental and biogeographic variables on alpha diversity of vascular plants.** Points represent the median relative importance of a predictor across 99 runs of a boosted regression tree model that jointly models vascular plant species richness in forest and non-forest formations. The bars connect the 2.5th and the 97.5th percentiles of the relative importance distribution across runs. The vertical dashed line separates variables with relative influence higher or lower than expected, i.e., those variables whose relative influence is higher or lower than 100% divided by the number of variables ($n = 20$).

(400 m²), the estimated alpha diversity of vascular plants (median prediction of each pixel of 2.5 arcminute resolution across the 99 resampled subsets) ranged from 1 to 120 species (median across all pixels = 22, interquartile range or IQR = 10; Fig. 1A, Supplementary Table 1). The areas with alpha diversity above the global 95th percentile (hereafter 'hotspots') were the forest-steppe region of easternmost Europe and Siberia, East Asia, Borneo and New Guinea, the eastern coast of Australia, the western Congo Basin, eastern Madagascar, the Andean-Amazonian foothills, the South American Atlantic Forest ('Mata Atlântica') and the Appalachian Mountains. Coldspots (i.e., areas with alpha diversity at a given grain size below the global 5th percentile) occurred in the Atlantic and Mediterranean part of Europe, central and western India, southern Australia, central Africa – specifically the eastern Guinean forest and the Sudanian savanna belt – and along the Pacific coast of North America. At the intermediate grain (1000 m²), the median estimated richness per grid cell in forest ecosystems ranged from 1 to 197 vascular plant species (global median across all grid cells = 29, IQR = 13) (Fig. 1B, Supplementary Table 1). Compared to the finest grain, all the hotspots in the equatorial region (Indonesia, Borneo, Andean-Amazonian foothills) increased in extent, whereas hotspots in the temperate and boreal regions either disappeared or shrank considerably. The coldspots in Western and Southern Europe and western North America remained, while those in central Africa diminished in size. Finally, at the coarsest grain (1 ha), average species richness per grid cell ranged from 2 to 921 species (median = 40, IQR = 39; Fig. 1C, Supplementary Table 1). At this grain, the well-known difference in species richness between the tropics and the boreal and temperate regions became apparent. The South American hotspots became connected, forming a belt spanning from the Andean-Amazonian foothills through the Chiquitano dry forest to the southern Pantanal and the Mata Atlântica regions. The hotspot in the western Congo Basin increased in size (Fig. 1C). The temperate region contained no hotspots at this grain. The coldspot in southern Australia expanded to the eastern coast, while the coldspot in central Africa disappeared. The uncertainty in alpha diversity estimates,

quantified as the ratio between IQR and median across the 99 resampled subsets, was highest in the boreal regions of Canada, Central and Eastern Siberia, the Amazon and Sundaland (Supplementary Fig. 2).

For non-forest ecosystems, we used an alternative set of grains: 10 m², 100 m² and 1000 m², to match the most frequently used plot sizes in our database. At the finest grain (10 m²), the median estimated alpha diversity across the 99 resampled subsets ranged from 0 to 68 vascular plant species (median across all grid cells = 14, IQR = 7; Fig. 2A, Supplementary Table 1). At this grain, non-forest hotspots were widely distributed across the forest-steppe region of easternmost Europe and Siberia, the central loess plateau of China, southern Eastern Australia, the Drakensberg region in South Africa, subtropical South America and eastern North America. Coldspots were widespread in southern Central Asia, central and northwestern Australia, the Sahel region of Africa and along the Pacific coast of South America. At the intermediate grain (100 m²), the median estimated species richness per grid cell ranged from 0 to 90 (median = 17, IQR = 9, Fig. 2B, Supplementary Table 1), and the distribution of hotspots and coldspots remained essentially unchanged compared to the finest grain. At the coarsest grain (1000 m²), the median estimated richness per grid cell ranged from 0 to 184 species (median = 23, IQR = 13, Fig. 2C, Supplementary Table 1). Except for the Loess plateau in China, hotspots were almost exclusively concentrated in subtropical regions at this scale, especially southeastern Australia, Madagascar, the Appalachian region, and the Pantanal and southern Cerrado in South America. The location of coldspots hardly changed compared to finer grains. The uncertainty in alpha diversity estimates was highest in northern Canada, the Tibetan Plateau and the Persian Gulf region (Supplementary Fig. 2). A map jointly showing alpha diversity of forest and non-forest ecosystems at 1000 m² grain is available in the supplementary material (Supplementary Fig. 4).

Overall, the models showed a relatively high predictive power (average over 99 resampling iterations: Pearson's $r = 0.49$), even after implementing a spatially constrained, block cross-validation[33] that accounted for the residual non-independence of training and test

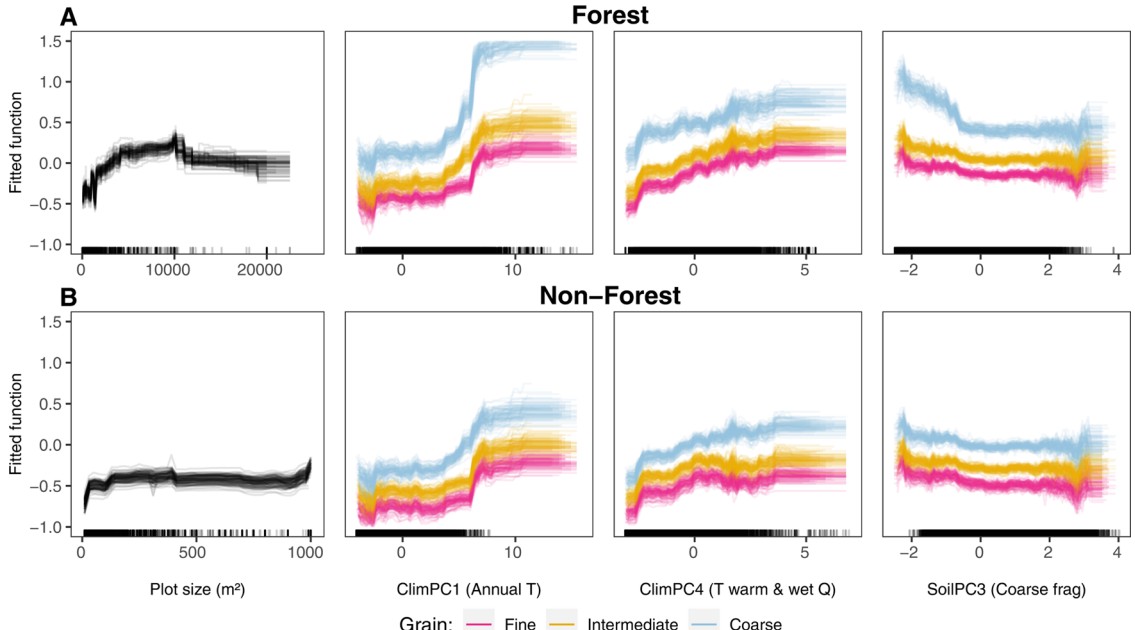

**Fig. 4 | Partial dependence plots for the main determinants of plant alpha diversity at different grain sizes.** These plots show the fitted function of the most influential variables explaining vascular plant alpha diversity at three different spatial grains, while holding all other predictor variables constant at their mean value. The fitted function is the difference between the response value at a given value of each predictor and the mean response value. Each line represents the fitted function for one of the 99 boosted regression tree model runs. Fine, intermediate and coarse grains correspond to 400 m², 1000 m² and 1 ha in forests (**A**) and 10, 100, and 1000 m² in non-forest ecosystems (**B**), respectively. Variables are sorted by decreasing relative influence. The rug plots on the *x*-axis display the distribution of the calibration data. Note the different range of plot sizes between forest and non-forest ecosystems.

datasets arising from the clustered nature of our database[34]. We found no major bias or trend in residuals across grain sizes, biomes or geographical regions (Supplementary Fig. 6), and the frequency distributions of observed and predicted values largely overlapped (Supplementary Fig. 7, Supplementary Table 2). The predicted values showed a slight tendency towards the sample mean with thinner tails at the extremes, which is a common feature of ensemble machine-learning methods, even with the bias-correction method we used (see Methods)[35]. Minor deviations only occurred for the dry mid-latitude and boreal biomes at coarse grains (Supplementary Fig. 7). Given the relatively small sample size for the wet tropics, we recommend interpreting the results for these regions with caution. For a complete description of model validation, see Supplementary Methods.

**Environmental and biogeographical determinants**
Our statistical models reveal which of the environmental and biogeographic variables tested appear to drive alpha diversity of vascular plants (Fig. 3). Among the predictors having a higher-than-expected relative influence, plot size, i.e., the grain size of the vegetation plot, consistently ranked first across the 99 resampled models. Climate also had a high relative influence in shaping alpha diversity patterns, especially annual mean temperature and the temperature of the warmest and wettest quarter of the year (PC1 and PC4, respectively, in a principal component analysis based on 18 bioclimatic variables). The ecoregional species pool, i.e., the estimated number of species occurring in the ecoregion in which a given plot is located[2], was the fourth most important predictor, highlighting the nested link between local and regional biodiversity. Finally, despite the expected importance of soil conditions for local plant diversity, only one soil variable, i.e., the percentage of coarse soil fragments, had an influence greater than 5%.

We created partial dependence plots to explore the directionality of these relationships and whether they are consistent across spatial grains and vegetation formations (Fig. 4). Plant alpha diversity increased non-linearly with increasing plot size. This effect saturated at relatively fine grains (~100 m²) in non-forest ecosystems and at 1 ha in forest ecosystems, which can be explained by the different grains at which forests and non-forests were sampled, and the different spatial structure of these vegetation types. Grain size interacted with most of the other predictors, as revealed by the different environment–richness relationships at different grains (Fig. 4). Alpha diversity increased when the size of the ecoregional species pool increased, but only for coarse grains. It also increased toward tropical regions (i.e., regions with higher temperatures of the warmest and wettest quarters, high scores on PC4) and at higher mean annual temperature (PC1), especially for coarse grains.

**Regional scaling anomalies in species richness across grain sizes**
Many areas with relatively high fine-grained alpha diversity also had high alpha diversity at coarser grains (Fig. 5). For forests, our models revealed consistently high alpha diversity across grains in Sundaland, the Congo Basin, Madagascar, as well as in the eastern Andean foothills, the Amazon Basin and the Southern American Mata Atlântica (Fig. 5A). Areas with consistently low alpha diversity across all grains were the western parts of the USA and Canada, the Atlantic region of Europe, Fennoscandia, the Mediterranean Basin, central and northern India, and southern Australia. However, not all areas with relatively high fine-grained richness also had high coarse-grained richness, and vice versa, revealing regional scaling anomalies in plant alpha diversity patterns[18]. Areas with high plant alpha diversity at coarse grains, but relatively low alpha diversity at fine grains, were the tropical forests of Africa and the Guiana Shield in South America. The opposite was true in the Eastern European forest–steppe belt, northeastern Argentina, Eastern Australia and New Zealand (Fig. 5A).

The regions hosting non-forest ecosystems with consistently high plant alpha diversity across grains were the European Alps, the forest-steppe of Eastern Europe and Siberia, the loess plateau of China,

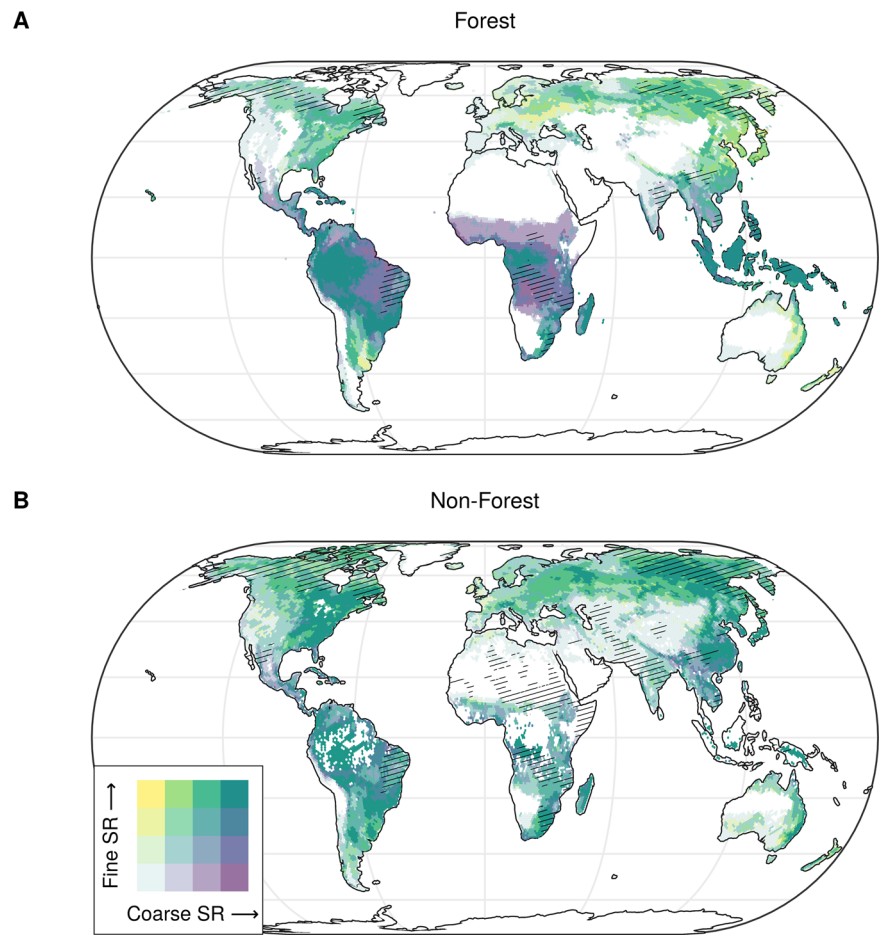

**A**  Forest

**B**  Non-Forest

**Fig. 5 | Regional scaling anomalies in species richness across grain sizes.** Correspondence between estimates of plant alpha diversity at fine and coarse grains for **A** forest and **B** non-forest ecosystems. Fine-grained alpha diversity was calculated at 400 m² and 10 m² for forest and non-forest ecosystems, respectively. Coarse-grained alpha diversity was calculated at 1 ha and 1000 m² for forest and non-forest ecosystems, respectively. We only show alpha diversity estimates where (**A**) forests would grow under current climate conditions and without human influence[208], or **B** the land cover 'herbaceous vegetation' occurs, based on a consensus map integrating multiple global remote sensing-derived land-cover products[209]. Color codes are based on quartile distributions of species richness at the two grains. Parallel hatching represents data-poor regions, i.e., regions farther than 500 km from any vegetation plots. Values are averaged over 7700 km² hexagons. SR: species richness. Source data are provided as a Source Data file.

Eastern Australia, eastern South Africa, Madagascar, the Chaco, Mata Atlântica and some other regions of South America, and eastern North America (Fig. 5B). Consistently low plant alpha diversity across grains occurred in Inner Asia and in the northern African desert and semi-desert regions, the Tibetan Plateau, Namib Desert, central Australia, the Atacama and High Monte deserts in the high Andean plateaus south of the equator as well as in the North American prairies and deserts. High coarse-grain species richness was associated with low fine-grain species richness in the Myanmar-Thailand-China borderland, Ethiopia and Mexico. The opposite situation was relatively rare, occurring locally in the temperate grasslands of southeastern Australia.

## Discussion

By simultaneously highlighting patterns at multiple spatial grains, our maps provide a nuanced picture of the pattern of alpha diversity of vascular plants. This complements our understanding of the distribution of biodiversity hotspots[36] and regional (i.e., gamma) vascular plant diversity[2–4,37]. Within the broad range of plot sizes commonly used for vegetation sampling, our maps distinguish between regions where high coarse-grained alpha diversity results largely from high fine-grained richness, and regions where high coarse-grained alpha diversity results more from species turnover between adjacent plant communities (i.e., fine-grained beta diversity).

Our results are consistent with previous studies suggesting that forests in Borneo, New Guinea, Madagascar, eastern South Africa and the Andean-Amazonian foothills are hotspots for plant biodiversity across all spatial grains[37]. There is considerable agreement between our map of 1-ha alpha diversity in forests and a recently published global map of tree species richness at the same grain[9]. Similarly, patterns of fine-grained alpha diversity in non-forest ecosystems are consistent with the local and regional patterns recently observed for alpine vegetation[38] and Palearctic grasslands[25]. We also found good agreement with previous research in the distribution of areas of low diversity (coldspots), such as the non-forest vegetation in the western Tibetan Plateau, the semi-desert regions of central Asia, coastal Somalia and the forests in the Pacific Northwest of North America, despite the large difference in grain[37].

In some regions, however, the difference between our results and previously reported patterns was striking. None of the regions holding the world records of plant alpha diversity appeared in our results[39]. The foothills of the Carpathians, for instance, are known for hosting semi-natural grasslands that are among the most species-rich plant communities globally at fine grains (e.g., >100 species in 16 m²)[39,40]. As many as 233 species (including 59 epiphyte species, not considered here) were observed in a 100 m² rain forest plot in Costa Rica[41]. At intermediate grains, very high plant species richness has been reported for the hemiboreal forests of the northern Russian Altai

(149 species per 1000 m²)[42] and Colombia (313 species per 1000 m²)[43]. At coarse grains, the world record is in Ecuador (942 species in 1 ha, including 172 epiphytes)[44]. Except for the Altai region, however, our maps do not show record high species richness in any of these regions. A general explanation is that our maps represent local averages across model runs, large areas (2.5 arcminute grid resolution) and a mixture of habitat types, so that the richest sites, which are rare in the landscape, have been averaged with neighboring sites that belong to other ecosystems with lower species richness. This is true, for instance, in Europe, where our data contained most non-forest vegetation types, including species-poor grasslands on acidic soils. The lack of data for epiphytes can partially explain why our model did not predict the expected high alpha diversity in Mesoamerica, where this growth form can account for up to 25% of forest species[41,45,46].

Interestingly, our models highlighted that alpha diversity does not differ markedly between temperate and tropical regions at the finest grains, but differences become more pronounced at coarser grains. This may reflect the often overlooked fact that tropical forests have a relatively species-poor herb layer compared to temperate forest ecosystems[46,47]. For instance, the high alpha diversity of trees in West African forests[7] is not accompanied by an equally high richness of herb or shrub species in the understorey. The low diversity in these understories could be due to the fact that tropical lowland forests have a closed canopy year-round[48], or that fires occur frequently, favoring grass-dominated, species-poor understories[49]. Together with the scarcity of data on epiphytes, a species-poor herb layer might explain why tropical lowland forests exhibit scaling anomalies, namely low alpha diversity at fine grains but high at coarse grains. If most of the diversity (or data) is in the tree layer, large vegetation plots are needed to ensure that the diversity of an ecosystem is appropriately sampled, as few tree individuals can physically co-occur at small sampling grains. We note, however, that uncertainties were high for tropical forests, requiring a cautious interpretation of these results.

In general, finding these scaling anomalies points to the role of beta diversity as a cross-scale diversity metric, and suggests that the relative contribution of different eco-evolutionary processes in determining plant diversity patterns varies between regions. In many tropical lowland forests, alpha diversity is low at fine grains but increases rapidly with increasing grain size. This is the case, for instance, in the western Amazon, where much of the regional (gamma) diversity depends on species turnover rather than on the coexistence of a high number of species at the same site[50]. This suggests that the tropics might be shaped by processes promoting species coexistence through a tighter packing in the niche space. Recent work found a latitudinal increase in niche specialization and marginality of trees towards the equators, which has been attributed to the stable climate and high productivity in the tropics[51]. Alternative explanations include rarity and priority effects related to high productivity[29], more uniform environmental conditions and stronger dispersal limitation at fine scales[28], or stronger mycorrhiza-mediated effects of interspecific competition and habitat adaptation[52] in the tropics compared to temperate regions. While the relative contribution of these processes remains a matter of speculation, our work points to the need for an improved understanding of the spatial variation of beta diversity in plant diversity analysis[53]. Beta diversity, rather than alpha diversity per se, appears to be the main driver of spatial differences in gamma diversity between temperate and tropical regions.

Conversely, we observed high plant alpha diversity at fine grains but relatively low alpha diversity at coarse grains in many temperate regions, including the Eastern European forest-steppe belt, East Asia and southeastern Australia. This pattern might be indicative of effective niche partitioning at fine grains and more homogeneous landscapes without dispersal barriers at coarse grains[54]. There is evidence that niche processes play a stronger role than neutral processes in determining fine-scale beta diversity at higher latitudes and

altitudes[28–30], where species are thought to have broader niches and be less responsive to geographical changes[55]. This is consistent with recent findings that the nestedness of tree communities increases with latitude, possibly due to the high share of ectomycorrhizal species in colder and wetter conditions[52]. Finally, high species richness at fine grains might also depend on plant size, as many small plants can coexist in a given grain size. Such conditions mainly occur in grasslands, e.g., in Eastern Australia, where this mechanism has been invoked to explain differences in beta diversity among vegetation types[56].

Our work allows us to rank the predictors of alpha diversity by their importance. Since the species–area relationship has often been described as one of the few rules in ecology[14], the high importance of plot size in our models is not surprising. Our important advance, however, is that by explicitly incorporating this nonlinear relationship into our models, we created a grain-independent model that links alpha diversity to multiple climatic, topographic and biogeographical predictors. We also showed that ecoregions with a large species pool are more likely to host species-rich communities. This pattern became disproportionately stronger at coarser grains, probably because at finer grains the maximum number of locally co-occurring species is constrained by the number of individuals that can fit into the grain. The other biogeographical covariates, namely biomes and realms, had very little effect on predicting alpha diversity. This is probably because they are closely related to other predictors with stronger explanatory power, i.e., macroclimate and ecoregions, respectively[3]. The increasing influence of macroclimate and ecoregional species pool with grain size is, however, in line with evidence on the role of climatic and geological histories of ecoregions on species pools[8,10,20,24]. This is not surprising since tectonic movements, uplift of mountain ranges, climatic stability, and glaciation events all play a role in driving regional speciation and extinction rates[3]. This result supports the view that, although intimately related, habitat filtering and biogeographical factors related to regional differences in geological and climatic history, have a different influence on patterns of alpha diversity at fine vs. coarse grains[10].

Although our study is based on the largest collection of global vegetation-plot data ever compiled, there are some shortcomings. The most important limitation is the uneven distribution of vegetation plots across biogeographical regions. Most of our data points were in Europe and other countries with a strong tradition of vegetation surveys, while the coverage of tropical areas, especially the Amazon and equatorial Africa, was poor (Supplementary Fig. 1). Furthermore, data from tropical forests were often incomplete, containing information on woody species only. Although the targeted search for additional data, coupled with the stratified resampling and statistical model we applied, mitigate these problems (see Methods), they clearly cannot compensate for the lack of comprehensive data on plant composition in many species-rich regions (especially large parts of the tropics). Ongoing initiatives to mobilize existing data, expand biodiversity surveys by including underreported growth forms such as herbs or epiphytes[45,46], and improve the overall taxonomic knowledge for these regions[47,57] are, therefore, high priorities in biodiversity research[58]. A second limitation is the scale mismatch between some very fine-grained vegetation plots and our use of coarse-grained environmental predictors, as highlighted by other global-scale biogeographical analyses[22]. Thus, our models ignore the mounting evidence of the strong modulating impact of local land cover, topographic heterogeneity and vegetation structure on climatic conditions, rendering the environmental conditions experienced by organisms at the local scale markedly different from those inferred from global macroclimatic models[59]. Finally, our analysis focuses on natural and semi-natural plant communities but ignores the role of human impacts and non-native species invasions. These effects are too diverse and multi-faceted to be included in a simple statistical model but clearly play a major role in the distribution of plant species, both at local and

regional scales[60]. Taken together, these limitations imply that although the accuracy of our models was relatively high, our results may still be missing important environmental drivers, especially at fine grain sizes.

Despite these limitations, our analysis provides important insights and is a step forward in mapping global plant diversity. First, it reinforces the idea that large-scale evolutionary and historical processes interact with local factors to shape plant communities[3,17,23]. Indeed, our models indicate that macroecological gradients have a consistent effect on plant alpha diversity, but with magnitudes that vary across grains. Second, by highlighting regional scaling anomalies in alpha diversity across different plot sizes, our study can improve our ability to predict biodiversity response to global change[11]. Third, our work adds a new dimension to our understanding of global biodiversity patterns and hotspots previously defined based on gamma diversity only. This could have implications for conservation. For example, coarse-grained hotspots might require networks of relatively large protected areas, whereas fine-grained hotspots might be more sensitive to biotic homogenization and more dependent on maintaining traditional management or a particular type of land use. Explicit consideration of the difference between coarse- and fine-grained hotspots complements the regional data on species richness and endemism commonly used for delineating global biodiversity hotspots.

## Methods

### Species richness data

The vegetation-plot database 'sPlot' (www.idiv.de/splot) collates 110 national or regional vegetation-plot datasets. Vegetation-plot records provide geo-referenced information on the presence and cover/ abundance of all vascular plants co-occurring within a delimited area. The sPlot database version 2.1 contains records from 1,121,244 vegetation plots surveyed between 1885 and 2015. These comprise 23,586,216 occurrence records for 58,066 vascular plant taxa, whose names have been standardized to a common nomenclature[15]. When the formation to which a plot belonged was not specified ($n = 137,146$ plots), we used the growth form of the recorded species[61] to classify a plot as forest or non-forest as in ref. 22. That is, we defined a plot record as forest if the sum of the cover values of all tree taxa was >25% of the sum of the cover values of all species in that plot, and as non-forest, if the sum of cover values of all low-growing taxa other than trees and shrubs was >90% of the sum of the cover values of all species in that plot. Plots not meeting either condition were excluded from the analysis, as well as all plots belonging to wetland or aquatic vegetation. Plots also had a wide variation in the sampled area (1–25,000 $m^2$). Therefore, we performed a preliminary screening and only retained plots sized between 100 and 25,000 $m^2$ for forest, and between 10 and 1500 $m^2$ for non-forest, as these are the most frequent plot sizes used by plant ecologists in the field. Plots without information on the sampled area were also excluded. Similarly, we excluded all plots that we could confidently assign to anthropogenic communities, here defined as any vegetation that is shaped by intensive and repeated human interference, including weed communities on arable land, ruderal vegetation and intensively managed pastures and meadows.

The data in the sPlot database are geographically biased since plots are unevenly distributed across geographical regions and formations (Supplementary Fig. 1), with relatively few data from the wet tropics. We therefore made a special effort to improve the data coverage in these regions by searching for publications and databases that report species richness, plot size and spatial coordinates of vegetation plots in the tropics. We focused on plots for which the full assemblage of vascular plants (with or without epiphytes) was sampled. However, such data were particularly scarce in many regions (e.g., the central Amazon, Western Ghats and Sundaland). For these regions, we also included data reporting woody species richness only (along with the diameter at breast height—DBH—used as the minimum sampling threshold). In total, we found information for an additional set of 1914 vegetation plots from 53 papers (Supplementary References). Of these, only 170 vegetation plots contained species richness information for all vascular plants. Finally, we scanned the Global Index of Vegetation-plot Databases[62] to retrieve additional datasets from the tropics, which were not included in sPlot 2.1. We obtained permission to use 11 local datasets, totaling 7929 additional vegetation plots (7385 with species richness data for all vascular plants). In total, our database contained 412,452 vegetation plots[41,59,63–196] (Supplementary Fig. 1, Supplementary Data 1).

### Data cleaning and geographical resampling

To further mitigate the remaining geographical bias in vegetation-plot distribution and to account for the fact that plot sizes vary markedly across regions and vegetation types (Supplementary Fig. 8), we applied a stratified resampling strategy that we repeated 99 times. We defined each stratum as a unique combination of realm[197], biome[15], broad formation (two classes: forest and non-forest), and plot size as a factor variable with four levels (small: ≤150, medium: 150–600, large: 600–1200, very large: >1200 $m^2$). These intervals were chosen to encompass the grains used for predictions (i.e., 10, 100, 400, 1000 and 1 ha, see below) while accounting for the fact that some plot sizes are more routinely used than others. For each stratum, we randomly sampled (without replacement) up to 100 vegetation plots in each iteration. If a stratum had fewer than 100 vegetation plots, we retained all of them. This procedure resulted in the selection of 17,972 plots in each iteration. The total number of plots used across the 99 iterations was 170,272. Altogether, these plots provided 9,953,940 occurrence records for 53,271 vascular plant taxa, i.e., ~15% of the estimated ~350,000 vascular plant species that exist. This figure is slightly underestimated, since for 1893 plots (59,299 occurrence records) only aggregated alpha diversity data were available, but no species-level data.

Not all vegetation plots were complete with respect to the sampled functional groups. Most records from tropical forest plots contained either only tree data, or only data on trees and shrubs (Supplementary Fig. 9). Excluding these plots would not be optimal, as it would have greatly reduced the spatial coverage of our dataset. Since most of these incompletely sampled plots were from the tropics, excluding them would also create the risk of introducing a strong spatial bias into our model. Therefore, we retained these plots in the dataset and included a new predictor variable called 'plants recorded' (three levels: 'complete vegetation', 'trees and shrubs only', 'trees only') in our statistical models (see below). Specifically, a plot belonged to the 'only trees' level if it only contained information on woody species with a diameter at breast height (DBH) larger than 5 cm. It belonged to the 'only trees and shrubs' level when it either contained information on all woody species (both trees and shrubs) but not herbs, or if the minimum DBH threshold used for sampling woody individuals was less than or equal to 5 cm.

As most of these incompletely sampled plots were in the tropics, we simulated the occurrence of incomplete plots also in the other biomes when resampling the full database. This was achieved by selecting some plots with complete vegetation information and recalculating their species richness when accounting for 'only trees', i.e., discarding all information on the occurrence of shrub and herb species, or for 'only trees and shrubs', i.e., discarding information on herbs. We limited this procedure to biomes with >10,000 plots with complete vegetation information (i.e., subtropics with winter rain, subtropics with year-round rain, temperate mid-latitudes). In these biomes, 20% of all the plots selected randomly within each resampling iteration (623 on average) were transformed this way. This corresponded to an increase in the number of incomplete plots in these selected biomes from 151 to 359 (on average over the 99 iterations), which is close to the average number of incomplete plots occurring in the other biomes ($n = 373$). By rarefying data to simulate

plots with incomplete vegetation records, we reduced the possible geographical bias resulting from the uneven distribution of incomplete plots across biomes. This allowed the use of incomplete plots from tropical regions (where complete plots are rare) when modeling the response of local vascular plant richness at the global scale (see below).

### Explanatory variables

Based on the plots' geographic coordinates, we retrieved bioclimatic, soil, topographic and biogeographical variables from external sources, which we used as explanatory variables for species richness modeling. We extracted all the 19 bioclimatic variables included in CHELSA v1.1[198], and seven soil variables at 250-m resolution from the SOILGRIDS project[199]. The soil variables were: (1) clay mass fraction (%); (2) silt mass fraction (%); (3) sand mass fraction (%); (4) coarse fragment fraction (%); (5) soil organic carbon content (g/kg); (6) soil pH (measured in water); and (7) cation exchange capacity. After standardizing and centering all 26 variables, we performed two principal component analyses (PCA), one for climate and one for soil. For subsequent analyses, we used the first five principal components for climate and the first four for soil, because these components accounted for more than 90% of the total variation in these ordinations. We interpreted these principal components based on the respective loadings of the corresponding environmental variables. For climate, the predictors with the highest loadings were: mean annual temperature for PC1; mean annual precipitation and mean diurnal temperature range for PC2; precipitation seasonality and precipitation of the wettest quarter for PC3; temperature of the wettest and temperature of the warmest quarter for PC4; and precipitation of the coldest quarter for PC5 (Supplementary Table 3, Supplementary Fig. 10). For soils, PC1 was mainly explained by soil bulk density; PC2 by sand content; PC3 by the percentage of coarse fragments and PC4 by soil pH (Supplementary Table 4, Supplementary Fig. 11).

To account for topographic heterogeneity, we also extracted data on plot topography from the EarthEnv.org data portal[200]. Specifically, we used terrain ruggedness (TRI, calculated at 50 km resolution), dominant landform (10 types at 1 km resolution: flat, peak, ridge, shoulder, spur, slope, hollow, footslope, valley, pit), and the number of landforms within a 50 km radius around each plot.

To account for historical and biogeographical factors, we included two predictors of the velocity of climate change between the Last Glacial Maximum and the present (one for temperature, one for precipitation) derived from ref. 201. These layers measure the local rate of displacement of climatic conditions and integrate macroclimatic shifts with local spatial topoclimatic gradients. Additionally, we considered two nominal biogeographical variables, realm[197] and biome[15], which we considered as rough proxies of the different geologic, biogeographical and climatic histories of different regions. The biomes were derived from Schultz's ecozones[202], which we modified to distinguish alpine areas[203]. Thus, our biomes are not nested within realms. As another surrogate for the biogeographical imprinting on alpha diversity patterns, we also accounted for regional effects by including the estimated size of the regional species pool for each of the 867 terrestrial ecoregions of the world[2].

We then considered three additional predictors: a binary variable distinguishing two broad formations (i.e., forest: True\False), a nominal predictor accounting for the different functional groups sampled in each plot (i.e., 'complete vegetation', 'only trees and shrubs' and 'only trees', see above), and plot size, i.e., the spatial grain used in vegetation sampling.

In total, we considered 20 predictors: five principal components summarizing climate, four principal components summarizing soils, three variables quantifying topographic heterogeneity, five related to biogeographical history, one representing vegetation formation and two related to sampling design. Multicollinearity among predictors

was limited, as no pair of predictors had Pearson's *r* coefficient greater than 0.64 (Supplementary Fig. 12).

### Statistical modeling

We used boosted regression trees (BRTs) to model the relationships between species richness and the explanatory variables. BRTs are nonparametric machine-learning models based on decision trees in a boosting framework. BRTs have few prior assumptions, are relatively robust against overfitting, missing data, and collinearity, and are very flexible in detecting nonlinear relationships and interactions among predictors[204]. We parameterized our BRTs as follows. We first set a tree complexity of 5 and a bag fraction of 0.5. We then systematically tested the combination between learning rates (from 0.00025 to 0.1) and the number of trees returning the highest 10-fold cross-validated model fit, using the *gbm.step* routine from the *dismo* package[205]. For each explanatory variable, we calculated its relative influence (i.e., the fraction of times a variable was selected for splitting a tree in each BRT model, weighted by the squared model improvement) across the 99 resampled sets. To visualize the relationship between species richness and the explanatory variables, we created partial dependence plots at selected grain sizes to visualize the marginal effect of a given predictor on the response variable. We considered an explanatory variable as relevant in the model if its relative influence (averaged over 99 resamplings) was greater than 5%, which is the expected share if all the 20 predictors had the same relative importance.

BRTs are unbiased on average, i.e., the sum of the residuals is close to zero. Yet, similarly to other ensemble machine-learning methods, they produce results that are biased in a different sense: small values are often overestimated and large values underestimated[35]. This happens because the final prediction is the unweighted average of a collection of regression trees, which inevitably leads to results biased towards the sample mean. To avoid this problem, we implemented a bias-correction algorithm called ROE: regression of observed on estimated values[35,206]. In the first step, we fitted a linear regression of the observed values on the fitted values:

$$S_{fit} = a + bS_{obs} \qquad (1)$$

where $S_{fit}$ is the vector of species richness predicted by a BRT in a given iteration, and $S_{obs}$ is the vector of observed species richness in that iteration. We then created a vector of bias-corrected, fitted species richness $S_{fit}^{bc}$ as:

$$S_{fit}^{bc} = \max\left[\frac{S_{fit} - a}{b}, 0\right] \qquad (2)$$

thus, introducing the constraint that $S_{fit}^{bc}$ is no smaller than zero[206].

We then used the above BRT models, together with the regression parameters *a* and *b*, to make bias-corrected predictions of local vascular plant richness at different plot sizes for all terrestrial pixels of the globe at 2.5 arcminute resolution. We did this separately for forest and non-forest ecosystems. For each pixel, we extracted the value for all 17 spatially explicit predictors (climate, soil, topography and biogeography) based on the pixel location. The variable 'forest' was set to 'True' for creating forest maps and 'False' for non-forest maps. For each of the 99 resampling iterations, we created multiple predictions, one for each selected sampling grain (i.e., 400 m², 1000 m² and 1 ha for forests, and 10, 100 and 1000 m² for non-forests). In all cases, we only predicted species richness for the complete vegetation (i.e., including trees, shrubs and herbs). We also mapped the variability of our predictions, as the interquartile range (IQR - i.e., the difference between the 75th and 25th percentiles) across the 99 resampling iterations. Finally, we created a map of ignorance[207] showing the geographic distance from the nearest vegetation plot used to calibrate our models (Supplementary Fig. 13). The map of

ignorance highlights the uncertainty due to the uneven geographic distribution of vegetation plots and shows areas with limited or no data where our estimation should be taken with caution. Based on this map, we highlighted all data-poor regions, i.e., regions located farther than 500 km from the nearest plot, by parallel hatching in our maps. Given the strong structural differences between forests and non-forest ecosystems, we presented the multi-grain maps of plant richness separately for these two broad formations in the main text. Nevertheless, we also produced a joint map at 1000 m² grain by complementing species richness estimates for forests with non-forest species richness for pixels outside the forest mask. For forests, we predicted all pixels where forests would grow under current climate conditions and without human influence[208]. For non-forest, we extracted all pixels where the land cover class 'herbaceous vegetation' occurs based on a consensus map integrating land-cover products derived from remote sensing[209].

### Model validation

We assessed model performance in three ways. First, we averaged the tenfold cross-validation across resampled sets obtained from the BRT output. Second, for each of the 99 resampled sets, we selected all plots not used in the specific set and calculated Pearson's correlation between species richness observed in a given plot and the respective BRT prediction at a grain corresponding to the plot area. As a third approach, we performed a spatially-constrained cross-validation[210]. We did this because our plots were spatially clustered and our spatial predictors had high spatial autocorrelation (2320 km on average across all the quantitative predictors, based on 5000 random samples, Supplementary Fig. 14). This means that even selecting plots completely independent of the training dataset does not ensure proper validation of our models, as the training and the test data remain spatially dependent. This violation of the fundamental assumption of model validation, namely the independence between training and test data, has been shown to affect many mapping models created with 'Big Data' approaches[34]. To avoid this problem, we divided the world into square spatial blocks whose size corresponds to the average spatial autocorrelation range of the quantitative predictors (i.e., 2320 km, $n = 84$, Supplementary Fig. 14). For each resampling, we randomly assigned each block to five folds using the function *spatialBlock* in the R package *blockCV*[33], which selects the most even spread of vegetation-plot data across folds in 99 iterations (Supplementary Fig. 15). We then refitted our BRT model five times for each resampling, each time using four out of five folds for training and the remaining fold for validation, and averaged Pearson's correlation coefficient between the observed and predicted species richness across folds. We also repeated this process separately for each biome separately, i.e., sequentially withholding all data located within a fold and a given biome for validation. We then reported the distribution of these correlation coefficients across the resampled sets, both when considering all plots, and when disaggregating by biomes. Finally, we checked the model residuals for spatial autocorrelation by fitting variogram models to the residuals using the function *variogram* from the R package *gstat*[211]. All analyses were performed in R 3.6.3[212]. Map boundaries derive from R package *rnaturalearth*[213].

### Reporting summary

Further information on research design is available in the Nature Research Reporting Summary linked to this article.

## Data availability

Source data are provided with this paper. All species richness data necessary to reproduce the results of this manuscript, including those retrieved through the literature search, and all raster files (format: GeoTiff) used to create the multi-grain maps of species richness are available at: https://doi.org/10.25829/idiv.3506-p4c0mo (ref. 214).

The vegetation-plot raw data contained in the sPlot database are available upon request by submitting a project proposal to sPlot's Steering Committee. The proposals should follow the Governance and Data Property Rules of the sPlot Working Group available on the sPlot website (www.idiv.de/splot). Source data are provided with this paper.

## Code availability

The code for reproducing the analyses presented in this article is available at: https://zenodo.org/badge/latestdoi/433417900 (ref. 215).

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

## Acknowledgements

The authors are grateful to thousands of vegetation scientists who sampled vegetation plots in the field or digitized them into regional, national or international databases. We appreciate the support of the German Research Foundation for funding sPlot as one of the iDiv research platforms (DFG FZT 118, 202548816). F.M.S. also acknowledges financial support within the funding programme Open Access Publishing by the German Research Foundation (DFG), and within the Rita-Levi Montalcini (2019) programme, funded by the Italian Ministry of University. In our analyses, we used the iDiv High-Performance Computing (HPC) cluster, for which we in particular acknowledge the support of Christian Krause. M.C. was supported by the Czech Science Foundation (project no. 19-28491X). V.D.P. was supported by the Brazilian National Research Council (CNPq grant 307689/2014-0). P.P. was supported by the long-term research development project No. RVO 67985939 of the Czech Academy of Sciences. A.P. gratefully acknowledges the Grant CONICYT PIA AFB170008 and Grant ANID PIA/BASAL FB210006. S.K.W. was supported by the Strategic Science Investment Fund of the NZ Ministry for Business, Employment and Innovation. A.G.-D.-M. acknowledges for the support of Agencia Española de Cooperación Internacional para el Desarrollo, Universidad CEU San Pablo (Madrid, Spain) and Universidad Privada Antonio Guillermo Urrelo (Cajamarca, Peru). H.K. acknowledges funding from the German Research Foundation (DFG) in the framework of the EFForTS project (Collaborative Research Centre 990. B.H. & J.K.N.'D. acknowledge the DynRecSe Project (C2D AMRUGE). E.A.-D. acknowledges the Red Col-Tree (Red de Monitoreo del bosque en Colombia). Ji.D. was funded by MŠMT Inter-excellence LTAUSA 18007, Czech Science Foundation (21-26883S). S.K. acknowledges funds by the 2019–2020 BiodivERsA joint call for research proposals, under the BiodivClim ERA-Net COFUND program (FeedBaCks, 193907). R.L.P. acknowledges funds by GeoPark Perú. J.J. acknowledges funds from various projects of Senacyt, Fonacon and Usac. O.P., A.M., and R.V. acknowledge support from the UK Department for International Development, a Research Fellowship to OP from the UK Natural Environment Research Council, the Mellon and MacArthur Foundations' support to the Missouri Botanical Garden, and the European Research Council for enabling our ForestPlots.net contribution to this project. C.V. was partly supported by the Fondation pour la Recherche sur la Biodiversité (FRB) and Electricité de France (EDF) in the context of the CESAB project 'Causes and consequences of functional rarity from local to global scales' (FREE). A full list of funding sources for each dataset listed in Supplementary Data 1 is available in Bruelheide et al.[15].

## Author contributions

F.M.S. and H.B. conceived the idea, with inputs by B.J.A. and U.J. F.M.S. performed the analysis. F.M.S. drafted the first version of the manuscript. B.J.A., H.B., U.J., M.C., V.D.P., R.F., S.H., M.K., J.L., F.S., Z.T., P.v.B. and S.K.W. provided substantial input on the manuscript and analytical framework. B.J.A., M.A.K., F.A, L.C., M.C., Jü.D., M.D.S., R.F., M.H., A.I., U.J., F.J., M.K., J.L., A.P., R.K.P., P.P., B.S., M.S., K.V., C.V., S.K.W., E.A.-D., P.D., Ji.D., B.H., A.G.-D.-M., J.J., S.K., S.K.-R., H.K., F.L., R.L.-P., J.K.N.'D., A.M.M., O.L.P., G.R.-T., P.S., K.S., B.J.S., R.V.M., H.-F.W., and K.W. provided parts of the data. All co-authors edited the manuscript and provided suggestions on how to improve the analyses.

## Funding

## Competing interests

The authors declare no competing interests.

## Additional information

¹German Centre for Integrative Biodiversity Research (iDiv), Halle-Jena-Leipzig, Puschstr. 4, 04103 Leipzig, Germany. ²Martin Luther University Halle-Wittenberg, Institute of Biology/Geobotany and Botanical Garden, Am Kirchtor 1, 06108 Halle, Saale, Germany. ³BIOME Lab, Department of Biological, Geological

and Environmental Sciences (BiGeA), Alma Mater Studiorum University of Bologna, Via Irnerio 42, 40126 Bologna, Italy. [4]Biodiversity Research Institute (CSIC/UO/PA), University of Oviedo, Campus de Mieres, Gonzalo Gutierrez Quiros, 33600 Mieres, Spain. [5]Masaryk University, Faculty of Science, Department of Botany and Zoology, Kotlářská 2, 611 37, Brno, Czech Republic. [6]University of Nottingham, School of Geography, University Park, NG7 2RD Nottingham, UK. [7]University of Zurich, Systematic and Evolutionary Botany, Zollikerstrasse 107, 8008 Zurich, Switzerland. [8]UMR CNRS 7058 "Ecologie et Dynamique des Systèmes Anthropisés" (EDYSAN), Université de Picardie Jules Verne, 1 Rue des Louvels, 80037 Amiens Cedex 1, France. [9]Manaaki Whenua - Landcare Research, Ecosystems and Conservation, 54 Gerald Street, 7608 Lincoln, New Zealand. [10]Shahjalal University of Science and Technology, Department of Forestry and Environmental Science, Akhalia, 3114 Sylhet, Bangladesh. [11]Sapienza University of Rome, Department of Environmental Biology, P.le Aldo Moro 5, 00185, Rome, Italy. [12]Universidad Rey Juan Carlos, Department of Biology and Geology, Physics and Inorganic Chemistry, c/ Tulipán s/n, 28933 Móstoles, Spain. [13]Zurich University of Applied Sciences (ZHAW), Vegetation Ecology Group, Institute of Natural Resource Sciences (IUNR), Grüentalstr. 14, 8820 Wädenswil, Switzerland. [14]University of Bayreuth, Plant Ecology, Bayreuth Center of Ecology and Environmental Research (BayCEER), Universitätsstr. 30, 95447 Bayreuth, Germany. [15]Wageningen University and Research, Environmental Sciences Group (ESG) Department, Plant Ecology and Nature conservation Group (PEN), Wageningen Campus, Building 100 (Lumen), P.O. Box Postbus 47, Droevendaalsesteeg 3, 6700 AA Wageningen, The Netherlands. [16]Tanta University, Faculty of Science, Botany & Microbiology Department, El-Geish st., Tanta University, 31527 Tanta, Egypt. [17]Transilvania University of Brasov, Department of Silviculture, Sirul Beethoven 1, 500123 Brasov, Romania. [18]University of Rostock, Faculty of Agricultural and Environmental Sciences, Justus-von-Liebig-Weg 6, 18059 Rostock, Germany. [19]Universidad de Concepción, Laboratorio de Invasiones Biológicas (LIB). Facultad de Ciencias Forestales, Victoria 631, 4030000 Concepción, Chile. [20]Instituto de Ecología y Biodiversidad (IEB), Las Palmeras 342, 7750000 Santiago, Chile. [21]University of North Carolina, Department of Biology, Campus Box 3280, 27599-3280 Chapel HIll, NC, USA. [22]Czech Academy of Sciences, Institute of Botany, Department of Vegetation Ecology, Zámek 1, 25243 Průhonice, Czech Republic. [23]Faculty of Environment UJEP, Pasteurova 3632/15, 400 96 Ústí nad Labem, Czech Republic . [24]Universidade Federal do Rio Grande do Sul, Department of Ecology, Av. Bento Gonçalves 9500, 91501-970 Porto Alegre, RS, Brazil. [25]Santa Clara University, Department of Biology, 500 El Camino Real, 95053 Santa Clara, CA, USA. [26]Palmengarten Frankfurt, Scientific Service, Siesmayerstr. 61, 60323 Frankfurt, Germany. [27]Senckenberg Biodiversity and Climate Research Centre, Data and Modelling Centre, Senckenberganlage 25, 60325 Frankfurt, Germany. [28]Peking University, College of Urban and Environmental Sciences, Yiheyuan Rd. 5, 100871 Beijing, China. [29]Institute of Environmental Sciences, Leiden University, 2333 CC Leiden, the Netherlands. [30]Institute of Biodiversity and Ecosystem Research, Department of Plant and Fungal Diversity and Resources, Acad. Georgi Bonchev St., bl. 23, 1113 Sofia, Bulgaria. [31]CEFE, Univ Montpellier, CNRS, EPHE, IRD, Montpellier, France. [32]Universidad Nacional Abierta y a Distancia, Escuela de Ciencias Agropecuarias y Ambientales, Sede Nacional, Cl. 14 Sur # 14-23, 111411 Bogotá, Colombia. [33]Sigur Nature Trust, Chadapatti, Mavinhalla PO, Nilgiris, 643223 Mavinhalla, India. [34]Department of Botany, Faculty of Science, University of South Bohemia, 370 05 České Budějovice, Czech Republic. [35]Cirad, UPR Forêts et Sociétés, Yamoussoukro, Côte d'Ivoire. [36]Université de Montpellier, UPR Forêts et Sociétés, Montpellier, France. [37]Institut National Polytechnique Félix Houphouët-Boigny, Département Forêts, Eaux, Environnement, Yamoussoukro, Côte d'Ivoire. [38]Universidad San Pablo-CEU, CEU Universities, Laboratorio de Botánica, Urbanización Montepríncipe, 28660 Boadilla del Monte, Spain. [39]Universidad de San Carlos de Guatemala, Escuela de Biología, Ciudad Universitaria, zona 12, 1012 Guatemala City, Guatemala. [40]University of Copenhagen, Department of Geosciences and Natural Resource Management, Rolighedsvej, 23, 2400 Copenhagen, Denmark. [41]University of Göttingen, Biodiversity, Macroecology & Biogeography, 37077 Göttingen, Germany. [42]University of Göttingen, Centre of Biodiversity and Sustainable Land Use (CBL), 37077 Göttingen, Germany. [43]Universidad de la República, Departamento de Sistemas ambientales, Facultad de Agronomía, Av. Garzón 780, 12900 Montevideo, Uruguay. [44]Smithsonian National Zoo and Conservation Biology Institute, Washington, DC, USA. [45]Universidad Nacional de San Antonio Abad del Cusco, Av. de la Cultura 733, Cusco, Peru. [46]Jardín Botánico de Missouri Oxapampa, Bolognesi Mz-E-6, Oxapampa, Pasco, Peru. [47]Université Félix Houphouët-Boigny, Laboratoire de Botanique, Campus de Cocody, Abdijan, Côte d'Ivoire. [48]University of Leeds, School of Geography, Woodhouse Lane, LS2 9JT Leeds, UK. [49]Estación de Biodiversidad Tiputini, Colegio de Ciencias Biológicas y Ambientales, Universidad San Francisco de Quito USFQ, Quito, Ecuador. [50]Charles University, Department of Botany, Benátská 2, 12801 Prague, Czech Republic. [51]INIBOMA (CONICET-UNCOMA), Department of Ecology, Pasaje Gutierrez 125, 8400 Bariloche, Argentina. [52]Namibia University of Science and Technlogy, Biodiversity Research Center, Faculty of Natural Resources and Spatial Sciences, 13 Jackson Kaujeua Street, 10005 Windhoek, Namibia. [53]College of Tropical Crops, Hainan University, Haikou 570228, China. [54]Botany Department, Senckenberg Museum of Natural History, Görlitz, PO Box 300 154, 02806 Görlitz, Germany. [55]International Institute Zittau, Technische Universität Dresden, Markt 23, 02763 Zittau, Germany. ✉e-mail: francescomaria.sabatini@unibo.it

