## [Peer Review File · Nature Communications]

Global patterns of vascular plant alpha diversityREVIEWER COMMENTS

Reviewer #1 (Remarks to the Author):

The paper by Sabatini et al. used an impressive vegetation plot database to predict species richness at different spatial grain based on environmental variables and a machine-learning algorithm. From these predictions the authors produce global maps of biodiversity patterns (hot- vs. cold-spots, scaling anomalies, etc.) that they use to challenge Meyers' biodiversity hotspots. Though I recognize that handling such a huge database is a challenge in itself, the paper unfortunately shows a number of fatal weaknesses to recommend publication.

First of all, the paper is mainly descriptive and lack inferences on how the observed patterns in different biomes or regions, for instance scaling anomalies, can help understanding the relative role of the different eco-evolutionary processes (dispersal, filtering and biotic interactions) in local structuring of plant communities. A common hypothesis is indeed that in contrasted bioclimates such as in mountainous regions, environmental filtering is the main driver of species assemblages, while in smoother bioclimates, like in lowland tropical forests, limited dispersal drives assemblages. Conversely, the niche multidimensionality hypothesis supports that fine-tuned adjustments to numerous niche dimensions maintain a high local diversity. What are the expectations of these hypotheses in terms of multi-scale structure of local species richness in different communities or regions? Moreover, species richness is a very poor diversity indicator, and accounting for abundance distributions and/or phylogenetic relationships between species in a local community would help discussing underlying hypotheses.

My second major criticism is on the lack of consideration of spatial autocorrelation in the predictive spatial model. A recent paper (<https://doi.org/10.1038/s41467-020-18321-y>) indeed demonstrated that even when neither trend nor bias is visible in the residuals, as it is the case here, spatial autocorrelation can make the assessment of model performance by a non-spatial cross-validation scheme largely overoptimistic. The problem is all the more deleterious when the sampling design is sparse and machine learning algorithm that tend to overfit local observations is used. In such cases, the predictors play just the role of spatial variables modelling the distribution of observations even in the absence of any causal relationships with the data. The only way to avoid such drawback is to consider spatial cross validation scheme, such as blocking for instance, that allows testing model performance on validation data spatially independent from the training data. It is not clear to me whether the stratified sampling used in the present paper helps limiting this bias. My feeling is no, but the methods are too succinctly presented to be sure. Anyway, the problem deserves consideration and the reasons why the model used could be immune from spatial autocorrelation artifacts must be discussed somewhere.

My last major concern is about the potential impact that could have on conservation policies the recommendations made on the basis of a so rough analysis. Just to take an example, the Western Ghats Biodiversity hotspot, in India, is one of the most endangered hotspots, with thousands of endemic plant species threatened by a long mounting anthropogenic pressure and an ever-increasing fragmentation of natural forests. From this study, the region is considered a biodiversity coldspot. This is correct that due to its biogeographic isolation, local species richness is less than it could be in similar bioclimates in Sri Lanka or Malaysia. However, the high environmental heterogeneity of this mountainous region makes it regionally hyperdiverse relatively to the area it covers. What would be the take home message cached by environmental planners reading that it is a biodiversity coldspot? It is of uttermost responsibility of scientists to not oversell simple results, even based on large datasets, and to conduct thorough analyses when the goal is to improve conservation priority-settings. As a matter of fact, I had a look to the data behind the Western Ghats projections. From what I could found, they are based on 6 plots concentrated around the city of Sirsi (c. 200 000

inhabitants) in the central Western Ghats area. I couldn't check the size and type of plant community as information is not given in appendix and forestplots.net is not accessible for unregistered users (I asked for a login authorization which I didn't receive before sending this review). Anyway, the plots are concentrated in a small area with regard to the extent of the Western Ghats hotspot, so that it is difficult to draw conclusions from a so sparse sampling (as it is also the case for most tropical regions in this study; see Fig. S1). For the Western Ghats example, several additional large databases of vegetation plots covering the entire region are publicly available as shows a simple Google search, and could have been used to complement the analysis. And I guess the same holds for most tropical regions.

Reviewer #2 (Remarks to the Author):

This well-written manuscript describes local-level plant species richness patterns at varying grain sizes based on an impressive global dataset of plot surveys worldwide. The analytical framework is generally well-designed, using resampling techniques to minimize biases in the data. The study presents maps of predicted vascular plant species richness based on a boosted regression model trained on the observed plot data. The authors highlighted regions of the world with consistent high species richness across grain sizes (hotspots), regions with consistent low species richness (coldspots), and regions with inconsistent species richness (anomalies). They contrasted these findings with the previous knowledge and attempted to explain the patterns in environmental and biogeographical predictors. This study contains novel results and should be of interest to many researchers in plant ecology and macroecology.

My main concern about this study is whether the sample for training the model covers well enough the natural variation of vascular plant species richness across biomes, particularly in the complex and species-rich tropical forests. I am concerned that the predictions for tropical forests with year-round rain might be particularly underestimated. For example, these tropical forests were estimated to have the same median species richness as temperate and boreal forests at the smallest grain. Comparing species richness among biomes and regions of the world is a focus of the study, and the authors base their comparison on the predicted values. If the model substantially underestimates species richness in tropical forests while accurately estimates richness in well-sampled biomes (e.g. temperate forests), inferred diversity patterns might deviate significantly from reality.

Despite the sampled data being largely incomplete for tropical forests and some world regions, predictions are made even for these poorly sampled biomes and regions. Samples seem to be missing or poorly represented in many parts of the tropical world, particularly in Africa, Southeast Asia and Brazil. Samples from tropical forests typically include only woody species, which may account for only 50% or less of the vascular plant species in the communities, while samples from temperate regions typically include a complete survey of the vascular plant species. The authors recognize that tropical forest surveys do not include epiphytes, a group that may account for >20% of the species richness. However, I wonder how well the samples capture other growth forms often not fully surveyed, such as herbs, pteridophytes, lianas, small palms, shrubs and small trees and treelets (i.e., those with dbh < 2.5cm). According to Spicer et al. (2020), herbs, epiphytes and lianas alone may account for 50% of the vascular plant species in tropical forests. From Figure S4, it seems that the samples from tropical rainforests are largely based on trees only or trees and shrubs, while other biomes have a complete sampling of the plant community. A visual inspection of the same figure suggests there are over 100 plots with 'complete' sampling for tropical rainforests (biome with the lowest number of complete plots) while around 100,000 plots for temperate forests. When you think about the variation in species richness across tropical forests in the entire world, >100 plots with "complete" data seems rather insufficient for generating predictions for the whole world.

I recognize the impressive effort in correcting for biases in the training data using statistical methods. Still, I am not convinced that it is enough (or even possible) to overcome the lack of information from tropical forests (and to account for the striking differences in sampling effort). The statistical correction with resampling methods are generally good approaches, but I guess it can only go so far as to correct biases.

I appreciate that the authors recognize noteworthy contrasts between predicted and observed species richness. As discussed in the paper, the fact that the predictions are local averages across large areas might explain why the maximum predicted values are well below the maximum observed values (although it is unclear how much of this difference can be attributed to the averaging process). However, that does not explain why the median and lowest values are well below expectation. It makes sense that tropical rainforests with >300 species per 0.1 ha are rare, but I find it hard to believe that the median species richness would be 40 at 0.1 ha (or 24 at 400 m²). From my experience in the field, these values would barely account for trees and shrubs in early successional or heavily disturbed communities (not accounting for other groups like herbs and lianas, which tend to be more dominant in disturbed sites). For old-growth tropical rain forests, the predicted number of species (for the entire vascular plant assemblage) would likely not even account for trees and shrubs with dbh > 2.5 cm (e.g. Linares-Palomino et al. 2008), and it probably falls well below the richness of tree species with dbh > 10 cm in richer regions (e.g. Martini et al. 2007).

The predictions from the model could be a better estimation of the reality, but the mentioned issues with the sampled data do not inspire confidence that this is the case. On the contrary, it inspires that the model is likely severely underestimating tropical forests' vascular species richness.

In the discussion, the authors mentioned that the relatively species-poor herb layer in the tropical forests could explain some tropical forest having low species richness compared to temperate forests. According to the cited reference, this comparison makes sense in per cent terms, so herbs make up 25% of the tropical forests' species richness while that contribution is 80% in the temperate forests (Spicer et al. 2020). It is unclear whether this per cent difference would account for a lower or similar absolute number of species in tropical forests compared to temperate and boreal forests.

It seems troubling that there is no information about human disturbance and the successional stage. It could be generally acceptable if the comparison is mostly between relatively old-growth forests, but if the sample includes early-mid successional and/or heavily disturbed forests, you need to control for these on your model (or remove them); otherwise, you may be comparing apples to oranges.

I agree with the authors that this study showcases insights that can be gained from analyzing this impressive dataset of plot data. Maybe the paper could focus more on this aspect. Because it showcases the use of a dataset that may become popular, perhaps it is even more critical to avoid predictions and inferences beyond what the sampled data safely supports. The described issues with the samples from tropical forests cast doubts on the reliability of the comparisons between tropical forests and others biomes. Thus, perhaps avoid making predictions for this biome is a sensible way forward. Predictions could be made only for biomes/regions with enough sampling, and this would serve well to showcase the current possibilities with the dataset.

Some questions and suggestions (that might help to build more confidence in the results):

- 1. Could you show summary statistics of the observed data? E.g., could you present a table similar to Table S2 based on the observed data?**
- 2. What is the frequency distribution of species richness in the observed data? Could you show frequency distribution plots with observed vs predicted species richness across**

biomes?

2.1. Does the range of observed values match the range of predicted values per biome/region? I wonder whether extrapolations beyond the observed range could be particularly unreliable.

3. Could you help the reader to make sense of unexpected predicted values? E.g., could you describe real-world examples of tropical forests with year-round rain on areas predicted to have 8-24 species at the 400m² scale or 10-40 at the 1000m²?

4. Why are the predictions so blend across regions at the smaller grains?

4.1. Could the larger number of plots from temperate regions be biasing the model to predict values similar to those observed in the temperate region even in other biomes/regions? Perhaps a way to investigate this would be to compare predictions from a model trained on data only from a specific biome vs a model trained on all data.

5. Given that the sample from the tropics seems the most troubling one, wouldn't it be better to remove the tropics from the model?

6. Given the model might be underestimating species richness in tropical forests, how fair it is to recommend the results to conservation prioritization (lines 432-434)?

References:

Linares-Palomino, R. et al. 2008. Non-woody life-form contribution to vascular plant species richness in a tropical American forest. - *Plant Ecology* 201: 87.

Martini, A. M. Z. et al. 2007. A hot-point within a hotspot: a high diversity site in Brazil's Atlantic Forest. - *Biodiversity and Conservation* 16: 3111-3128.

Spicer, M. E. et al. 2020. Seeing beyond the trees: a comparison of tropical and temperate plant growth forms and their vertical distribution. - *Ecology* 101: e02974.

Reviewer #3 (Remarks to the Author):

Review of "Global patterns of local plant diversity" by Sabatini et al.

This paper represents a very impressive analysis of an enormous body of data, representing an aspect of plant diversity for which such compilations are only now permitting large-scale analyses like these. The authors ought to be congratulated for their extremely impressive efforts and the huge amount of work, and data, that this represents. It deserves to be published and I would be happy to see this manuscript in print and would certainly want to cite it in my own work. In general it is very well written and the analyses appear to be robust and to have been sensibly undertaken. However, as is inevitable, there are several issues that I think ought to be addressed by the authors before publication, in order to improve the quality of their manuscript.

Firstly, I think it is very misleading to claim, as is done in the Summary Paragraph (line 104) and implied in the Materials and Methods (line 552), that the analysis is based on 1.1 million georeferenced local plant assemblages. It is not; over half the plots were excluded from the analysis (Material and Methods, line 565). It therefore seems irrelevant that there might be some "23,586,216 occurrence records for 58,066 vascular plant taxa" (lines 553-554) if most of these are excluded. 58,066 species is something like ~15% of global vascular plant species, so presumably the analysis presented here is based on <10% of vascular plant species. There are understandable reasons why this is so, but the relevant metric that should be presented for the reader would be the number of occurrence records and species actually represented in the data that were used in the analysis, not the total number in the whole database when most of these were not used. This ought to be explicitly stated in the Materials and Methods instead of the total number of records and species in the whole database, and will help the reader to set the work presented here into an appropriate broader context.

Secondly, the authors have called their manuscript "Global patterns of local plant

diversity", when by local they mean alpha (α -) diversity ("local (i.e. alpha-) plant diversity" Main text, line 127). However, they are not completely explicit about this and for the rest of the paper use the term "local" as synonymous with α -diversity, whereas beta (β -) diversity is also an important component of diversity and can certainly vary over short spatial distances (within the scale of the grains they use for analysis, for example) and therefore also be 'local'. I think the introduction to the main text needs an explicit sentence somewhere along the lines of "this paper presents an analysis of global patterns of plant alpha diversity, from vegetation plots", to make this clear to the reader. As they later acknowledge, "species richness is only one facet of biodiversity" (Main text, line 383) and the role of beta (β -) diversity is mentioned ("In the case of western Amazonia, much of the regional (=gamma) diversity probably depends on species turnover (=beta-diversity)", lines 384-385), but otherwise the whole focus of the paper is on α -diversity, which to my mind is not quite the same as 'local' diversity (see below). It might be when compared with regional gamma (γ -) diversity, but not much is made of this comparison (see also below). I think it would help with the interpretability of the paper if the authors were either explicit about meaning α -diversity when they say 'local', or use the term α -diversity instead of always using 'local'.

Thirdly, notwithstanding the size of the total data and the immense amount of work it has taken to compile and curate not only the individual datasets and to combine them into a single, federated database, there are clearly major biases within the data, particularly towards less diverse temperate ecosystems, especially within Europe, and away from more diverse tropical environments. The authors freely acknowledge this and have taken a series of sensible, pragmatic steps to try and ameliorate these biases. I have no particular issue with these decisions or with any other aspect of the methodology. One step that I was unable to follow, however, was the production of the global-scale maps that are the main explanatory figures (1 & 2) in the manuscript. Lines 665-667 state that "We then used the BRT models above to predict local-scale species richness of complete vegetation assemblages (i.e., including trees, shrubs and herbs) at different plot sizes, and created a global map having 0.1° resolution", but I was left none the wiser about exactly how this was done. It ought to be clear to the reader but currently it is not at all. I also did not understand why the scales presented in the different maps (forest: 400 m², 1,000 m², and 1 ha, Main text, line 170; non-forest: 10 m², 100 m² and 1,000 m², Main text, line 211) do not relate at all to the size thresholds used in the stratification (small: ≤150, medium: 150-600, large: 600-1200, very large: >1200 m², Material and Methods, line 573). I apologise for my lack of understanding and there may be perfectly sensible reasons for this but if so I was left unaware of them as the manuscript does not mention this, when it ought to be explicitly stated. I understand the pragmatic reason for treating forest and non-forest plots differently (trees are bigger, and forest plots often only record trees) but, leading on from this issue, Figure 5 shows multi-scale 'forest' versus 'non-forest' plots, which is a different way of presenting the same information already given in Figures 1 and 2. Examining Figure 5A and Figure 5B, clearly many pixels/hexagons contain both forest and non-forest vegetation (unsurprisingly). As the grains used for forest and non-forest plots overlap, I feel it would be more informative for the reader to have three panels: forest data across scales (Fig. 5A); non-forest data across scales (Fig. 5B); and combined forest and non-forest data at 1000 m². Otherwise, there is no figure in which forest and non-forest data are both presented together, even though they were seemingly analysed jointly (Main text, line 167) and overlap each other in grain.

Fourthly, in their analysis of factors promoting high local species richness (α -diversity) presented in Figure 3, while the separation of related abiotic variables through ordination appears sensible, I think some of their biotic variables remain non-independent. Quite a bit of to-ing and fro-ing between the Materials and Methods and Figure 3 is needed for this so I can't be completely sure, but I assume 'Ecoregion species pool' (second-most important variable) is the numeric estimate of total number of species in the ecoregion from Kier et al. (2005) (reference 11), but this number of species is also going to partly be a product of the categorical variables Realm (sixth-most important) and Biome (15th-most). The binary 'forest/non-forest' variable is also

included as well as Biome, although presumably whether or not a particular vegetation plot is from a forest or not is also reflected in which biome it is in. These are relatively minor issues but a little more discussion of these factors would be welcome, if the word count allows it. It is no great surprise that plot size is the factor with the largest effect on species richness, or that the size of the regional species pool also has a large effect. However, I was disappointed that the Discussion for this paper did not really discuss these results in more depth.

Fifthly, I don't know if the authors could perform some more sensitivity analyses for their results to explore other spatial scales and species components, but it seems that some of their results could still be artefacts of the analysis. For me, the two main findings from this research are that: fine local-scale α -diversity can be much more similar between tropical and temperate regions than differences in regional γ -diversity would suggest; but that while some areas showed consistent patterns of local species richness across grains, other areas did not. Therefore, to what extent might it be the case that areas differing in local diversity patterns across scales are caused by the particular grain sizes used or the missing species components in the data? For example, perhaps not having major species components such as epiphytes or herb layers in forest plots could explain consistent scale-independent diversity patterns, or conversely inconsistent scale-dependent diversity patterns, or might analysing species richness at still larger scales show a different pattern to those at smaller scales in either case? There are clearly very strong relationships between local α -diversity and regional γ -diversity patterns, but I was disappointed that the Discussion does not go into any real detail on these (despite citing several studies of regional γ -diversity patterns in the Introduction). The obvious missing piece of the picture, though, that is barely addressed in this paper, is clearly β -diversity, and how this varies spatially. Beyond citing a single study, (Condit, R. et al. Beta-Diversity in Tropical Forest Trees. *Science* 295, 666-669) as an explanation for diversity patterns in a single region (western Amazonia), β -diversity is hardly mentioned in the Discussion at all. As spatial scales increase and local species richness tends towards regional species richness, clearly this is due to the effect of β -diversity bringing more new species into the orbit of local species estimates. So the issue for me is less about 'how does α -diversity vary spatially?' but more about 'why does β -diversity vary so much between the tropics and the temperate regions?', and 'how and why does it also vary within both temperate and tropical regions?', because if local α -diversity is mostly the product of plot size and regional species pool, then β -diversity is what is causing the differences between local α -diversity and regional γ -diversity. I realise that the data the authors use for this study does not lend itself to analyses of β -diversity, but even if they cannot analyse this issue themselves it seems obtuse to barely even refer to this phenomenon in the Discussion. Word limits are always an issue in papers like these, but this seems such a glaring oversight that some less-essential text could surely be sacrificed for better discussion of the role of β -diversity: it is almost as if the wrong question has been asked. Partly this is itself a reflection of the paper not really being completely explicit that it only attempts to present and explain patterns of plant α -diversity (as opposed to 'local' diversity), which therefore makes the omission of β -diversity seem all the more glaring.

In summary therefore, I feel the paper would benefit very much from: stating exactly how many records and species were used in the analysis; being clear that by 'local' diversity the authors are confining themselves to analysing α -diversity patterns; stating clearly how the maps were derived, the spatial thresholds defined and presenting combined forest and non-forest results together somewhere; elaborating on the probable impact of missing species components, and the relationship between α -diversity and β -diversity, on the outcome of their analyses; and adding some additional sensitivity analyses and greater discussion of the relationship between α -diversity and γ -diversity patterns. If these issues could be addressed I would be very happy to see this paper published in *Nature Communications*.

REVIEWER COMMENTS

Reviewer #1 (Remarks to the Author):

The paper by Sabatini et al. used an impressive vegetation plot database to predict species richness at different spatial grain based on environmental variables and a machine-learning algorithm. From these predictions the authors produce global maps of biodiversity patterns (hot- vs. cold-spots, scaling anomalies, etc.) that they use to challenge Meyers' biodiversity hotspots. Though I recognize that handling such a huge database is a challenge in itself, the paper unfortunately shows a number of fatal weaknesses to recommend publication.

Response: We did not intend our results to challenge Myers' hotspots. We analyse local species richness (=alpha diversity) at three grains, but our largest grains of analysis (1 ha for forests and 0.1 ha for non-forests) are still small relative to the resolution of most previous global analyses. We focus on species richness, but not on endemism, and we do not look at threats to species or habitats, so the results are not comparable to hotspots like those of Norman Myers. Particularly in tropical and sub-tropical mountainous regions, for example, high levels of endemism are expected, which means that 'coldspots' of local species richness in our analysis may be hotspots in Myers' framework, without contradiction. We do attempt to use our data and methods to reveal how local species richness may change across the globe and how this metric compares (or not) to biodiversity hotspots commonly used for setting conservation goals. Because the two are measuring different things, it is not an attempt to challenge Myers' hotspots, but instead to provide a different viewpoint, which has not been addressed before (see next response). In the new version, we tried to clarify that our results are meant to complement, rather than challenge previous knowledge on biodiversity hotspots (L126-128, L369-372, 490-492)

First of all, the paper is mainly descriptive and lack inferences on how the observed patterns in different biomes or regions, for instance scaling anomalies, can help understanding the relative role of the different eco-evolutionary processes (dispersal, filtering and biotic interactions) in local structuring of plant communities. A common hypothesis is indeed that in contrasted bioclimates such as in mountainous regions, environmental filtering is the main driver of species assemblages, while in smoother bioclimates, like in lowland tropical forests, limited dispersal drives assemblages. Conversely, the niche multidimensionality hypothesis supports that fine-tuned adjustments to numerous niche dimensions maintain a high local diversity. What are the expectations of these hypotheses in terms of multi-scale structure of local species richness in different communities or regions? Moreover, species richness is a very poor diversity indicator, and accounting for abundance distributions and/or phylogenetic relationships between species in a local community would help discussing underlying hypotheses.

Response: We appreciate that our dataset enables many different analyses, and the ones suggested by the referee would make for a nice paper, or set of papers. However, that research is different to what we have attempted here, which is well described at the start of referee 2's review. Ours is, to our knowledge, the first attempt to map the species richness of locally co-occurring species at the global scale (as opposed to using checklists or range maps, which do not inform about which species occur together on the ground). Rather than trying to incorporate a whole research programme into one paper, we have tried to stay

focused on what we think is a necessary place to start from: to produce global maps at a range of (local) grain sizes, and examine the differences between the grains, which are relevant for developing our understanding of the nature of global biodiversity patterns, and for on-the-ground conservation. Our study does not address community assembly, but instead the major patterns and drivers of species richness that co-occur locally (i.e. within the same vegetation plot) as a result of assembly process. A logical next step could be to use the data to test competing hypotheses relating to these processes, which is what the referee appears to be suggesting that we should do.

Similarly, although we recognise that species richness can be criticised as a diversity measure for some types of analysis, it is a measure that both has value in itself and has a long history of research interest - attested by the many thousands of papers that focus on it, including very many current ones.

To clarify these points, we have added some text to illustrate how knowledge on local species richness (alpha diversity) patterns at different scales can shed light on the underlying mechanisms (L162-164): *“For example, the discrepancies between species richness patterns at different grains might point to regional or biome-related variation in the roles of habitat heterogeneity, dispersal barriers or environmental filtering[27].”*

Again in L435-440: *“How local species richness scales with increasing grain size, however, depends on the dominant eco-evolutionary processes structuring local plant communities. In the case of western Amazonia, for instance, much of the regional (gamma) diversity depends on species turnover (beta-diversity), rather than on the coexistence of a high number of species at the same site[50]. This points to more uniform environmental conditions at fine scales and dispersal limitation at larger scales. Such conditions are also known from regions that have been geologically and climatically stable for a long time, such as Western Australia[51]. In contrast, comparably high plant species richness at fine grains but low plant species richness at coarse grains might be an indicator of effective niche partitioning at fine scales and more homogeneous landscapes without dispersal barriers at coarse scales[52]. High species richness at fine scales also depends on plant size, as many small plants can coexist in a given grain size. Such conditions mainly occur in grasslands, e.g. in Eastern Australia.”*

See also L441-454: *“Yet, this effect became disproportionately stronger at coarser grains, probably because at finer grains the maximum number of locally co-occurring species is constrained by the number of individuals that fit into the grain. The other biogeographical covariates, namely biomes and realms, played a very marginal effect at predicting local species richness. Probably, this is because they are closely related to other predictors with stronger effects, i.e., macroclimate and ecoregions, respectively[3]. The increasing influence of macroclimate and ecoregional species pool with grain size is, however, in line with evidence on the role of climatic and geological histories of ecoregions on species pools [8,10,20,24]. This is not surprising, since tectonic movements, uplift of mountain ranges, climatic stability, and glaciation events all play a role in driving regional speciation and extinction rates[3]. This result supports the view that, although intimately related, habitat filtering and biogeographical factors related to regional differences in the geological and climatic history, have a different influence on patterns of species richness at fine vs. coarse grains[10].”*

My second major criticism is on the lack of consideration of spatial autocorrelation in the predictive spatial model. A recent paper (<https://doi.org/10.1038/s41467-020-18321-y>) indeed demonstrated that even when neither trend nor bias is visible in the residuals, as it is the case here, spatial autocorrelation can make the assessment of model performance by a non-spatial cross-validation scheme largely overoptimistic. The problem is all the more deleterious when the sampling design is sparse and machine learning algorithm that tend to overfit local observations is used. In such cases, the predictors play just the role of spatial variables modelling the distribution of observations even in the absence of any causal relationships with the data. The only way to avoid such drawback is to consider spatial cross validation scheme, such as blocking for instance, that allows testing model performance on validation data spatially independent from the training data. It is not clear to me whether the stratified sampling used in the present paper helps limiting this bias. My feeling is no, but the methods are too succinctly presented to be sure. Anyway, the problem deserves consideration and the reasons why the model used could be immune from spatial autocorrelation artifacts must be discussed somewhere.

Response: Thanks for this important and insightful comment. We followed the suggestion and recomputed our model validation using spatial blocks. As expected (for the reasons you alluded to), the correlation between observed and predicted species richness is less when using spatial blocks than when using randomly chosen points in cross-validation: across the 99 resampling iterations, Pearson's correlation was, on average, 0.49 compared with the 0.79 obtained from randomly sampled points. While this difference between the two validation methods is substantial, as it always is for this sort of datasets, the underlying signal that it reveals is still strong - far from the situation of the examples cited in the paper you referenced (Ploton et al., 2021), in which the spatial cross-validation rendered the models uninformative.

In the process of revising our work according to the above suggestions, we paid more attention to the spatial autocorrelation itself. The input covariates were, as expected, spatially structured (with an autocorrelation range of ~2320 km, on average). We used this autocorrelation range to define the size of our spatial blocks, as suggested by the authors of the R package blockCV (Valavi et al. 2018). The residuals of our BRT models showed no remaining spatial autocorrelation signal at all. We included a graph showing this absence of spatial autocorrelation signals in model residuals for resampling iteration 1 (Supplementary Figure 15).

We expanded the method section to clearly show how we approached cross-validation based on the new approach. Please see L684-705. Please also note that the code is available for referees to scrutinise, which can be another way of confirming the quality and transparency of our methods.

My last major concern is about the potential impact that could have on conservation policies the recommendations made on the basis of a so rough analysis. Just to take an example, the Western Ghats Biodiversity hotspot, in India, is one of the most endangered hotspots, with thousands of endemic plant species threatened by a long mounting anthropogenic pressure and an ever-increasing fragmentation of natural forests. From this study, the region is considered a biodiversity coldspot. This is correct that due to its biogeographic isolation,

local species richness is less than it could be in similar bioclimates in Sri Lanka or Malaysia. However, the high environmental heterogeneity of this mountainous region makes it regionally hyperdiverse relative to the area it covers. What would be the take home message cached by environmental planners reading that it is a biodiversity coldspot? It is of uttermost responsibility of scientists to not oversell simple results, even based on large datasets, and to conduct thorough analyses when the goal is to improve conservation priority-settings. As a matter of fact, I had a look to the data behind the Western Ghats projections. From what I could find, they are based on 6 plots concentrated around the city of Sirsi (c. 200 000 inhabitants) in the central Western Ghats area. I couldn't check the size and type of plant community as information is not given in appendix and forestplots.net is not accessible for unregistered users (I asked for a login authorization which I didn't receive before sending this review). Anyway, the plots are concentrated in a small area with regard to the extent of the Western Ghats hotspot, so that it is difficult to draw conclusions from a so sparse sampling (as it is also the case for most tropical regions in this study; see Fig. S1). For the Western Ghats example, several additional large databases of vegetation plots covering the entire region are publicly available as shows a simple Google search, and could have been used to complement the analysis. And I guess the same holds for most tropical regions.

Response: As we stated above, our results on hotspots and coldspots should be interpreted in terms of local (alpha) species richness only, while global biodiversity hotspots are also based on endemic richness (mostly estimated at regional scale) and threat levels. Rather than challenging Myers' definition of hotspots, therefore, our work adds a new dimension to our understanding of the distribution of vascular plants on Earth, i.e., how these are assembled in local communities to compose more or less species-rich communities at the local scale, and how this richness varies across local (!) spatial grains. Hence, a hotspot according to Myers' definition can also be a local coldspot according to our work and definition, without challenging or altering the concept of regional hotspots. Rather our new dimension of local hotspots/coldspots sheds new lights on the underlying processes behind regional hotspots that are local coldspots, as it suggests beta diversity to be the main driving force behind regional hotspots. As such, we do not claim that our results should be directly used to amend conservation priorities, nor to prioritize conservation areas. Yet, our work clearly has some conservation implications, as coarse-grained hotspots might be targeted with different actions than fine-grained hotspots. In the new version of our manuscript, we try to clarify these aspects, and dispel any possible confusion about our intentions.

We deleted the reference to conservation implications in the abstract and main text and toned it down in the discussion section (L485-492): *“Third, our work adds a new dimension to our understanding of global biodiversity patterns and hotspots, and this may have conservation implications. For example, coarse-grained hotspots might require networks of relatively large protected areas, whereas fine-grained hotspots might be more sensitive to biotic homogenization and might more strongly depend on maintaining the traditional management or a particular type of land use. Explicitly accounting for the difference between coarse- and fine-grained hotspots complements the regional data on species richness and endemism commonly used for delineating global biodiversity hotspots.”*

Similarly, we rephrased the opening paragraph of the discussion to make clear that our results complement rather than challenge Myer's hotspots and previous knowledge on the global distribution of vascular plant diversity (L370-372).

With regard to Western Ghats, please note that the new version of our work does not suggest forests in the Western Ghats to be a coldspot of alpha-diversity any more (while this seems to be the case for other regions in central and northern India). Yet, we concede that data for this area were indeed sparse in the first version of the ms. To address this issue, we compiled additional data from tropical and subtropical regions, performing a comprehensive survey to include as many papers or datasets as we could find (see response to reviewer 2 for details). Western Ghats was one of the main targets of our search for more data, and we managed to find about 10,000 new vegetation plots, 186 of which are from Western Ghats and were compiled/contributed by a local researcher (Priya Davidar), who is now part of our co-author team, or stemmed from a published data paper (Ramesh et al. 2010 – Ecology). Unfortunately, the large database contained in the “Biological Information System of India” (which we believe is the database reviewer 1 was referring to) is not functional anymore, as neither the download function nor the spatial viewer work on our browser. The contact email listed on the website is also not working, and Prof. Kushwaha, who was the corresponding author of a 2012 paper describing the database (Roy P, Kushwaha S, Roy A. 2012. Landscape level biodiversity databases in India: status and the scope. PROCEEDINGS OF THE NATIONAL ACADEMY OF SCIENCES, INDIA SECTION B: BIOLOGICAL SCIENCES 82:261-269) kindly answered to our request to access the data stating that the project is closed and data are not retrievable at the time being. Additional search using the Global Index of Vegetation Databases (GIVD.info) and Google Scholar, returned few additional data, as most of the published papers for the area do not report open-access data.

Reviewer #2 (Remarks to the Author):

This well-written manuscript describes local-level plant species richness patterns at varying grain sizes based on an impressive global dataset of plot surveys worldwide. The analytical framework is generally well-designed, using resampling techniques to minimize biases in the data. The study presents maps of predicted vascular plant species richness based on a boosted regression model trained on the observed plot data. The authors highlighted regions of the world with consistent high species richness across grain sizes (hotspots), regions with consistent low species richness (coldspots), and regions with inconsistent species richness (anomalies). They contrasted these findings with the previous knowledge and attempted to explain the patterns in environmental and biogeographical predictors. This study contains novel results and should be of interest to many researchers in plant ecology and macroecology.

Response: This paragraph is an excellent summary of what we did and conveys very well the general aim of our work. We wish to thank reviewer 2 for appreciating our work and, even more, for the useful insights on how to improve it.

My main concern about this study is whether the sample for training the model covers well enough the natural variation of vascular plant species richness across biomes, particularly in the complex and species-rich tropical forests. I am concerned that the predictions for tropical forests with year-round rain might be particularly underestimated. For example,

these tropical forests were estimated to have the same median species richness as temperate and boreal forests at the smallest grain. Comparing species richness among biomes and regions of the world is a focus of the study, and the authors base their comparison on the predicted values. If the model substantially underestimates species richness in tropical forests while accurately estimates richness in well-sampled biomes (e.g. temperate forests), inferred diversity patterns might deviate significantly from reality.

Response: We agree with this criticism, and indeed we had intense discussions among the authors about these issues before the original submission, resulting in many of the model checking and validation procedures that were already in place in that first submission. Clearly, however, we were not fully successful in addressing it in the first submission, and have tried our best to amend the manuscript, particularly with regard to coverage of species-rich tropical forests. In the new version of the ms, we did three things to mitigate / solve this issue of data scarcity in the tropics:

1. We compiled additional data from tropical regions, performing a comprehensive survey to compile data from as many papers or datasets as we could find;
2. We implemented a bias-correction algorithm to adjust for our species richness predictions;
3. We changed the way we show and discuss the results for the wet tropics, to highlight the high uncertainties in the predictions for this biome.

For an extensive description of the additional data we collected (9843 plots in the tropics), please see our response to the next comment. Here we focus on points 2 and 3.

With respect to point 2: Our relatively low species richness predictions for the tropics might depend on an intrinsic weakness of ensemble machine-learning methods, which can result in systematic bias including underestimation of extremely high values. We certainly underestimated this effect in our first submission. All ensemble machine learning regression models such as BRTs can be prone to a subtle form of bias. While the sum of the residuals (observed minus predicted values) is generally unbiased and close to zero, these methods tend to overestimate small values and underestimate large values. This might explain the fact that areas expected to support extremely species-rich communities, did not emerge. We now used a bias-correction algorithm, which we only became aware of when seeking to address this criticism, to mitigate this problem. We explain this procedure at L639-653.

The scatterplot between observed and predicted values (Supplementary Figure 4) shows that bias-corrected residuals are considerably closer to the 1:1 line now. Even if some bias remains, this is a feature common to most bias-correction methods (see for instance Xu et al., 2015, Carbon Balance Management, Zhou et al., 2016, Fish & Fisheries; Belitz & Stackelberg, 2021). This is mentioned in L264-269.

With respect to point 3: Even after re-fitting the models with additional data and including the bias-correction algorithm, we must acknowledge that tropical regions were still undersampled. Thus, we decided to highlight the uncertainties that remained for the tropics in a better way, both graphically and in the text. Graphically, we did this by enlarging the areas covered with hatches, which now refer to areas more than 500 km (instead of 1000 km) from the closest vegetation plots, and in the text by better describing the data

limitations, warning the reader that these results should be taken with caution both in the results (L268-270) and discussion section (L420-422). *“We note, however, that uncertainties for tropical forests were high, which calls for a cautious interpretation of these results.”*

Despite the sampled data being largely incomplete for tropical forests and some world regions, predictions are made even for these poorly sampled biomes and regions. Samples seem to be missing or poorly represented in many parts of the tropical world, particularly in Africa, Southeast Asia and Brazil. Samples from tropical forests typically include only woody species, which may account for only 50% or less of the vascular plant species in the communities, while samples from temperate regions typically include a complete survey of the vascular plant species. The authors recognize that tropical forest surveys do not include epiphytes, a group that may account for >20% of the species richness. However, I wonder how well the samples capture other growth forms often not fully surveyed, such as herbs, pteridophytes, lianas, small palms, shrubs and small trees and treelets (i.e., those with dbh < 2.5cm). According to Spicer et al. (2020), herbs, epiphytes and lianas alone may account for 50% of the vascular plant species in tropical forests. From Supplementary Figure 4, it seems that the samples from tropical rainforests are largely based on trees only or trees and shrubs, while other biomes have a complete sampling of the plant community. A visual inspection of the same Supplementary Figure suggests there are over 100 plots with 'complete' sampling for tropical rainforests (biome with the lowest number of complete plots) while around 100,000 plots for temperate forests. When you think about the variation in species richness across tropical forests in the entire world, >100 plots with "complete" data seems rather insufficient for generating predictions for the whole world.

I recognize the impressive effort in correcting for biases in the training data using statistical methods. Still, I am not convinced that it is enough (or even possible) to overcome the lack of information from tropical forests (and to account for the striking differences in sampling effort). The statistical correction with resampling methods are generally good approaches, but I guess it can only go so far as to correct biases.

Response: The lack of 'complete' vegetation surveys for the tropics is, unfortunately, a real issue, as the referee correctly pointed out. When revising our manuscript, we decided to better scan the scientific literature, and used our network of contacts to further improve the data coverage for tropical regions. We managed to find a lot more data from the Tropics (nearly 10,000 additional vegetation plots), which is a very substantial improvement compared to the former version of our work. We describe what we did in the methods, and provide a full list of the new data sources in the Supplementary References.

Please see L516 and following: “Data in sPlot are geographically biased since plots are unevenly distributed across geographical regions and formations (Supplementary Figure 1), with relatively few data from the wet tropics. We thus put special effort into improving the data coverage in these regions, looking for papers and databases reporting species richness, plot size, and spatial coordinates for vegetation plots in the tropics for which the full assemblage of vascular plants (with or without epiphytes) was sampled. These data were, however, particularly scarce in many regions (e.g., central Amazon, Western Ghats, Sundaland). For these regions, we also included data reporting woody species richness only (together with the diameter at breast height – DBH – used as minimum sampling threshold).

In total, we found information for an additional set of 1,914 vegetation plots from 53 papers (Supplementary references). Only 170 vegetation plots contained species richness information for all vascular plants. Finally, we scanned the Global Index of Vegetation-plot Databases[58] to retrieve additional datasets from the tropics, which were not included in sPlot 2.1. We secured permission to use 11 local datasets, totalling 7,929 additional vegetation plots (7,385 with species richness data for all vascular plants). In total, our database contained 417,180 vegetation plots (Supplementary Figure 1, Supplementary Table 1, Supplementary References)."

Clearly, compared to the temperate biomes, the tropics are still undersampled. But this is exactly the reason why we resampled all data 100 times, stratifying by biomes, and plot size, before fitting our models. Any statistical approach can only go so far in attempting to correct biases, unfortunately. Therefore, even if we believe the new version of our manuscript provides much sounder and convincing results for the tropics, we have ensured that we are now more cautious when discussing our results for the tropics, as described above.

I appreciate that the authors recognize noteworthy contrasts between predicted and observed species richness. As discussed in the paper, the fact that the predictions are local averages across large areas might explain why the maximum predicted values are well below the maximum observed values (although it is unclear how much of this difference can be attributed to the averaging process). However, that does not explain why the median and lowest values are well below expectation. It makes sense that tropical rainforests with >300 species per 0.1 ha are rare, but I find it hard to believe that the median species richness would be 40 at 0.1 ha (or 24 at 400 m²). From my experience in the field, these values would barely account for trees and shrubs in early successional or heavily disturbed communities (not accounting for other groups like herbs and lianas, which tend to be more dominant in disturbed sites). For old-growth tropical rain forests, the predicted number of species (for the entire vascular plant assemblage) would likely not even account for trees and shrubs with dbh > 2.5 cm (e.g. Linares-Palomino et al. 2008), and it probably falls well below the richness of tree species with dbh > 10 cm in richer regions (e.g. Martini et al. 2007). The predictions from the model could be a better estimation of the reality, but the mentioned issues with the sampled data do not inspire confidence that this is the case. On the contrary, it inspires that the model is likely severely underestimating tropical forests' vascular species richness.

Response: Please see our responses to the comments above for more on this. After refitting the models with the new data from the tropics, and including a bias-correction algorithm, our estimations for the tropics are now much more realistic. Even if the new global medians for the wet tropics (Supplementary Table 2) for 0.1 ha forests increased only marginally (new median = 46), the range (and IQR) are now much higher with maximum predicted values as high as 186 species. These figures are better in line with the observed values (Supplementary Table 3): at the 0.1 ha grain, for the set of 322 plots with complete vegetation, the median is 27 species, and the maximum is 307. To convince the reader about the soundness of our results, we followed your suggestions (below) and created a graph showing the distribution of observed and predicted values, when disaggregating the data based on (1) vegetation type, (2) biome, and (3) sampling completeness (Supplementary Figure 5). We discuss these results in L261: "We found no major bias or trend in the residuals across grain sizes, biomes or geographical regions (Supplementary Figure 4), and the frequency distributions of

observed and predicted values were mostly overlapping (Supplementary Figure 5, Supplementary Table 3). Predicted values showed a slight tendency towards the sample mean with thinner tails of extreme values, which is a common feature of ensemble machine-learning methods even with the bias-correction method we set in place (see Materials and Methods)[34]. Minor deviations only occurred for the dry mid-latitude and boreal biomes at coarse grains (Supplementary Figure 5). Given the relatively low sample size for the wet tropics, we recommend interpreting the results for these regions with caution.”

One last note of caution: Comparing our predictions to species richness estimates obtained using spatially discontinuous plots (e.g., Gentry’s explored quadrat method, such as in Martini et al., 2007, referenced above) is not appropriate. The number of species captured using Gentry’s explored quadrat method will always be (considerably) higher, due to increase and inflation in beta-diversity, than when considering a contiguous plot of the same size (as we did here). Not to mention Gentry’s tendency to bend the rules of randomization in order to sample particular plants as reported in Phillips, O. and Miller, J.S. (eds) (2002) *Global patterns of plant diversity: Alwyn H. Gentry’s forest transect data set*. Missouri Botanical Garden Press, Missouri, USA.)

In the discussion, the authors mentioned that the relatively species-poor herb layer in the tropical forests could explain some tropical forest having low species richness compared to temperate forests. According to the cited reference, this comparison makes sense in per cent terms, so herbs make up 25% of the tropical forests' species richness while that contribution is 80% in the temperate forests (Spicer et al. 2020). It is unclear whether this per cent difference would account for a lower or similar absolute number of species in tropical forests compared to temperate and boreal forests.

Response: The new results clearly show that even if the proportion of species in the herb layer is much lower in tropical forests, these are still more species-rich than temperate forests, at least at coarse grains. We touch upon this point in L416: “Together with the scarcity of data on epiphytes, a species-poor herb layer might explain why lowland tropical forests exhibited scaling anomalies, where species richness was low at fine grains but high at coarse grains. If most of the diversity (or data) is in the tree layer, large vegetation plots are needed to ensure the diversity of an ecosystem is appropriately sampled, as only few tree individuals can physically co-occur at small sampling grains.”

It seems troubling that there is no information about human disturbance and the successional stage. It could be generally acceptable if the comparison is mostly between relatively old-growth forests, but if the sample includes early-mid successional and/or heavily disturbed forests, you need to control for these on your model (or remove them); otherwise, you may be comparing apples to oranges.

Response: We do not have accurate information on the disturbance level for most of our plots. However, the great majority of the datasets composing our database focus on natural and semi-natural vegetation, which exclude the most anthropogenically disturbed communities (agricultural fields and edges, planted meadows, ruderal communities). This is especially true for those datasets in the tropics and subtropics, which mostly stem from small research projects focusing on natural vegetation in relatively intact areas. Only some of

the largest datasets from Europe, which come from nation-wide archives covering all kind of vegetation types, sometimes include anthropic vegetation. In the revised ms, we now excluded all those vegetation plots that we could confidently attribute to anthropogenic communities. Please see L512: *'Similarly, we excluded all plots that we could confidently attribute to anthropogenic communities, here defined as any vegetation that is shaped by intensive and repeated human interference, including arable field and ruderal communities as well as fertilized and intensively utilized pastures and meadows.'* This led us to exclude an additional 23,437 plots from our initial selection.

I agree with the authors that this study showcases insights that can be gained from analyzing this impressive dataset of plot data. Maybe the paper could focus more on this aspect. Because it showcases the use of a dataset that may become popular, perhaps it is even more critical to avoid predictions and inferences beyond what the sampled data safely supports. The described issues with the samples from tropical forests cast doubts on the reliability of the comparisons between tropical forests and others biomes. Thus, perhaps avoid making predictions for this biome is a sensible way forward. Predictions could be made only for biomes/regions with enough sampling, and this would serve well to showcase the current possibilities with the dataset.

Response: These are well-made points. We still think it makes sense to make predictions for tropical areas, since we improved data coverage from the tropics (even if the uncertainty remains large). But we do agree that it is crucial to be transparent on the limitations on one's work, so that the reader can make an informed judgment. Please see our response to the points above, on how we (1) improved the data coverage for the tropical regions, (2) incorporated bias correction to reduce systematic underprediction of high values in our models, (3) cautioned the reader on the reliability of our results for these regions, and (4) discuss the limitations of our work, including toning down our description of the conservation implications that should be drawn from our work. We hope that this same referee will be able to review our revised manuscript, and judge the extent to which we have succeeded in addressing the important concerns he/she raised.

Some questions and suggestions (that might help to build more confidence in the results):

1. Could you show summary statistics of the observed data? E.g., could you present a table similar to Supplementary Table 2 based on the observed data?

Response: We added this information in Supplementary Table 3

2. What is the frequency distribution of species richness in the observed data? Could you show frequency distribution plots with observed vs predicted species richness across biomes?

Response: Good suggestion! We created a panel showing the distribution of observed and predicted species richness across plot sizes, sampling completeness, vegetation types (forest, non-forest), and biomes. See Supplementary Figure 5 and L261.

2.1. Does the range of observed values match the range of predicted values per biome/region? I wonder whether extrapolations beyond the observed range could be

particularly unreliable.

Response: As you can see in Supplementary Figure 5, the distributions of observed and predicted values are remarkably similar, with the only exception being the 'Dry Mid Latitude' biome, where predictions appear to be too narrowly distributed around the median.

3. Could you help the reader to make sense of unexpected predicted values? E.g., could you describe real-world examples of tropical forests with year-round rain on areas predicted to have 8-24 species at the 400m² scale or 10-40 at the 1000m²?

Response: We added some text in L261, and provided more information on the observed values in Supplementary Figure 5 and Supplementary Table 3

4. Why are the predictions so blend across regions at the smaller grains?

Response: It probably depends on the color scale (which is log-transformed). We have now stretched the scale to increase the contrast.

4.1. Could the larger number of plots from temperate regions be biasing the model to predict values similar to those observed in the temperate region even in other biomes/regions?

Perhaps a way to investigate this would be to compare predictions from a model trained on data only from a specific biome vs a model trained on all data.

Response: The resampling strategy we put in place is meant to prevent this problem. In each resampling iteration, we capped the number of plots from different biomes to be used in the respective BRT model. Comparing predictions across models trained on different data sounds like a sensible approach. However, we believe it is more informative and robust to compare predictions based on models which exclude one biome at the time. Also based on the suggestions from referee 1, we now implemented this approach when performing model validation. Besides including spatial blocks (see response to referee 1), we also validated our model using combinations of spatial blocks AND biomes. We explain our strategy in L697:

"We then refitted our BRT model five times for each resampling, each time using four out of five folds for training and the remaining fold for validation, and averaged the Pearson's correlation coefficient between the observed and predicted species richness across folds. We repeated this process also considering each biome separately, i.e., sequentially withholding all data located within a fold and within a given biome for validation."

The results of this new validation approach are shown in the updated version of Supplementary Figure 4. The variation across biomes is substantial (Pearson's r 0.25-0.75), but varies around the general mean of the cross-validation based on spatial blocks (see Supplementary methods).

5. Given that the sample from the tropics seems the most troubling one, wouldn't it be better to remove the tropics from the model?

Response: Please, see our responses to the comments above. In short, we prefer to keep the tropics in our analysis, and have instead improved their representation and analysis, and added more caution to our inference for that important part of the world.

6. Given the model might be underestimating species richness in tropical forests, how fair it is to recommend the results to conservation prioritization (lines 432-434)?

Response: This is fair point. We still believe that some of the implications of our work might be relevant for conservation, but we agree it might be dangerous to use our results to prioritize conservation. We decided, therefore, to tone down our description of conservation recommendations. Please see our response to the fourth comment by referee 1, above, for more details.

References:

- Linares-Palomino, R. et al. 2008. Non-woody life-form contribution to vascular plant species richness in a tropical American forest. - *Plant Ecology* 201: 87.
- Martini, A. M. Z. et al. 2007. A hot-point within a hotspot: a high diversity site in Brazil's Atlantic Forest. - *Biodiversity and Conservation* 16: 3111-3128.
- Spicer, M. E. et al. 2020. Seeing beyond the trees: a comparison of tropical and temperate plant growth forms and their vertical distribution. - *Ecology* 101: e02974.

Reviewer #3 (Remarks to the Author):

Review of "Global patterns of local plant diversity" by Sabatini et al.

This paper represents a very impressive analysis of an enormous body of data, representing an aspect of plant diversity for which such compilations are only now permitting large-scale analyses like these. The authors ought to be congratulated for their extremely impressive efforts and the huge amount of work, and data, that this represents. It deserves to be published and I would be happy to see this manuscript in print and would certainly want to cite it in my own work. In general it is very well written and the analyses appear to be robust and to have been sensibly undertaken. However, as is inevitable, there are several issues that I think ought to be addressed by the authors before publication, in order to improve the quality of their manuscript.

Response: We thank reviewer 3 for appreciating our work and, even more, for the useful insights on how to improve it.

Firstly, I think it is very misleading to claim, as is done in the Summary Paragraph (line 104) and implied in the Materials and Methods (line 552), that the analysis is based on 1.1 million georeferenced local plant assemblages. It is not; over half the plots were excluded from the analysis (Material and Methods, line 565). It therefore seems irrelevant that there might be some "23,586,216 occurrence records for 58,066 vascular plant taxa" (lines 553-554) if most of these are excluded. 58,066 species is something like ~15% of global vascular plant species, so presumably the analysis presented here is based on <10% of vascular plant species. There are understandable reasons why this is so, but the relevant metric that should be presented

for the reader would be the number of occurrence records and species actually represented in the data that were used in the analysis, not the total number in the whole database when most of these were not used. This ought to be explicitly stated in the Materials and Methods instead of the total number of records and species in the whole database, and will help the reader to set the work presented here into an appropriate broader context.

Response: We broadly agree with this critique, and we now only report the number of plots effectively used in our analysis in the summary paragraph. Note however that, given our geographically stratified resampling approach, the species coverage of the data we used in the final analysis is actually much higher than you suggest. We now clarify this both in the summary paragraph and methods in L543: "This procedure resulted in the selection of 17,972 plots in each iteration. The total number of plots used across the 99 iterations was 170,700. Altogether, these plots provided 9,743,939 occurrence records for 50,290 vascular plant taxa, i.e., 87% of the species contained in sPlot 2.1."

Secondly, the authors have called their manuscript "Global patterns of local plant diversity", when by local they mean alpha (α -) diversity ("local (i.e. alpha-) plant diversity" Main text, line 127). However, they are not completely explicit about this and for the rest of the paper use the term "local" as synonymous with α -diversity, whereas beta (β -) diversity is also an important component of diversity and can certainly vary over short spatial distances (within the scale of the grains they use for analysis, for example) and therefore also be 'local'. I think the introduction to the main text needs an explicit sentence somewhere along the lines of "this paper presents an analysis of global patterns of plant alpha diversity, from vegetation plots", to make this clear to the reader. As they later acknowledge, "species richness is only one facet of biodiversity" (Main text, line 383) and the role of beta (β -) diversity is mentioned ("In the case of western Amazonia, much of the regional (=gamma) diversity probably depends on species turnover (=beta-diversity)", lines 384-385), but otherwise the whole focus of the paper is on α -diversity, which to my mind is not quite the same as 'local' diversity (see below). It might be when compared with regional gamma (γ -) diversity, but not much is made of this comparison (see also below). I think it would help with the interpretability of the paper if the authors were either explicit about meaning α -diversity when they say 'local', or use the term α -diversity instead of always using 'local'.

Response: Thank you for raising this important point. We now state explicitly that our analysis focusses on local species richness (i.e. alpha diversity) only, and unified the terminology throughout the text. We believe the term "local species richness" is more accurate than alpha-diversity for two reasons. First, in our paper we consider different grains, each one characterized by a value for alpha diversity. Giving them all the same name "alpha diversity" does not do justice to the fact that the differences between grains are possibly (partly) caused by beta diversity. In other words, the concept of alpha-diversity is not really scalable across grains, which is an additional reason to consider grain explicitly, rather than focussing on descriptors of diversity that can be interpreted in multiple ways and have no direct link to grain size. For additional details on beta diversity, please see our response to point 4, below. Second, we believe 'local species richness' is a more readily understandable terminology, especially for a general science journal such as Nature Communications, where some readers might not be familiar with the alpha\beta\gamma framework. We refer to this reasoning in L166: "We refer to local species richness, rather than alpha diversity, because the latter concept is usually used for a single grain size only."

Thirdly, notwithstanding the size of the total data and the immense amount of work it has taken to compile and curate not only the individual datasets and to combine them into a single, federated database, there are clearly major biases within the data, particularly towards less diverse temperate ecosystems, especially within Europe, and away from more diverse tropical environments. The authors freely acknowledge this and have taken a series of sensible, pragmatic steps to try and ameliorate these biases. I have no particular issue with these decisions or with any other aspect of the methodology. One step that I was unable to follow, however, was the production of the global-scale maps that are the main explanatory figures (1 & 2) in the manuscript. Lines 665-667 state that “We then used the BRT models above to predict local-scale species richness of complete vegetation assemblages (i.e., including trees, shrubs and herbs) at different plot sizes, and created a global map having 0.1° resolution”, but I was left none the wiser about exactly how this was done. It ought to be clear to the reader but currently it is not at all.

Response: We have clarified our approach in the methods section. Please see L654: “We then used the BRT models above, together with the regression parameters a and b , to make bias-corrected predictions of local vascular plant richness at different plot sizes for all terrestrial pixels of the globe at a 2.5 arcminute resolution. We did this separately for forest and non-forest ecosystems. For each pixel, we extracted the value for all 17 spatially-explicit predictors (climate, soil, topography and biogeography) based on the pixel location. The variable ‘forest’ was set to TRUE when creating forest maps, or FALSE for non-forest maps. For each of the 99 resampling iterations, we created multiple predictions, one for each selected sampling grain (i.e., 400, 1,000 and 10,000 m² for forests, and 10, 100 and 1,000 m² for non-forests). In all cases, we only predicted species richness for the complete vegetation (i.e., including trees, shrubs and herbs).”

Please note that we erroneously specified our maps had a 0.1° resolution in the first version of the ms. The correct resolution is 2.5 arcminutes and we have corrected the text accordingly.

I also did not understand why the scales presented in the different maps (forest: 400 m², 1,000 m², and 1 ha, Main text, line 170; non-forest: 10 m², 100 m² and 1,000 m², Main text, line 211) do not relate at all to the size thresholds used in the stratification (small: ≤150, medium: 150-600, large: 600-1200, very large: >1200 m², Material and Methods, line 573). I apologise for my lack of understanding and there may be perfectly sensible reasons for this but if so I was left unaware of them as the manuscript does not mention this, when it ought to be explicitly stated.

Response: These intervals were chosen to include the plot sizes used in the main text as their (more or less) central points, while accounting for the fact that some plot sizes are more often used by ecologists than others. For instance, a size of 400 m² is more commonly used to sample temperate forest vegetation than a size of 500 m², because the first can be easily achieved using a 20x20m quadrat. For this reason, we believe that focusing on plots 150-600 m² can provide a better approximation of the species richness expected to be found at 400 m², than using an interval such as 101-400 m². We now made our reasoning clearer in the text (L539-541).

I understand the pragmatic reason for treating forest and non-forest plots differently (trees are bigger, and forest plots often only record trees) but, leading on from this issue, Figure 5 shows multi-scale 'forest' versus 'non-forest' plots, which is a different way of presenting the same information already given in Figures 1 and 2. Examining Figure 5A and Figure 5B, clearly many pixels/hexagons contain both forest and non-forest vegetation (unsurprisingly). As the grains used for forest and non-forest plots overlap, I feel it would be more informative for the reader to have three panels: forest data across scales (Fig. 5A); non-forest data across scales (Fig. 5B); and combined forest and non-forest data at 1000 m². Otherwise, there is no figure in which forest and non-forest data are both presented together, even though they were seemingly analysed jointly (Main text, line 167) and overlap each other in grain.

Response: Thank you for this suggestion. We now have created a 'joint panel' showing the species richness for both vegetation types. For those pixels where both forest and non-forest occur, we show the species richness of forest, rather than non-forest ecosystems, since in the majority of cases forest represents the potential natural vegetation. Please see Supplementary Figure 3 and description in L255-257.

Fourthly, in their analysis of factors promoting high local species richness (α -diversity) presented in Figure 3, while the separation of related abiotic variables through ordination appears sensible, I think some of their biotic variables remain non-independent. Quite a bit of to-ing and fro-ing between the Materials and Methods and Figure 3 is needed for this so I can't be completely sure, but I assume 'Ecoregion species pool' (second-most important variable) is the numeric estimate of total number of species in the ecoregion from Kier et al. (2005) (reference 11), but this number of species is also going to partly be a product of the categorical variables Realm (sixth-most important) and Biome (15th-most). The binary 'forest/non-forest' variable is also included as well as Biome, although presumably whether or not a particular vegetation plot is from a forest or not is also reflected in which biome it is in. These are relatively minor issues but a little more discussion of these factors would be welcome, if the word count allows it. It is no great surprise that plot size is the factor with the largest effect on species richness, or that the size of the regional species pool also has a large effect. However, I was disappointed that the Discussion for this paper did not really discuss these results in more depth.

Response: This is a fair point. While we fully agree that part of the variation explained by biomes might be shared with the other variables you mentioned (realm, ecoregional species pool), we would like to note that our definition of biome was derived from Schultz's (2005) ecozones, which we modified to also include alpine areas. As such, our biomes are not simply nested within realms, nor nested within ecoregions. We preferred, therefore, to keep the predictors in our model, as this is consistent with the stratified resampling of the data, and with the spatial block cross-validation of the model output. We now mention the low explanatory power of this variable, though (L443-446).

With regard to plot size, we now restructured the discussion to highlight the pervasive role of spatial grain when modelling species richness (L435): *"Our work enables the ranking of predictors of local species richness by their importance. As the species-area relationship has often been described as one of the few rules in ecology[14], the high importance of plot size in our models is unsurprising. Our important advance, however, is that by explicitly*

incorporating this non-linear relationship into our models, we created a grain-independent model linking species richness to multiple climatic, topographic and biogeographical predictors. We also showed that ecoregions with a large species pool are more likely to host species-rich communities. Yet, this effect became disproportionately stronger at coarser grains, probably because at finer grains the maximum number of locally co-occurring species is constrained by the number of individuals that fit into the grain. The other biogeographical covariates, namely biomes and realms, played a very marginal effect at predicting local species richness. Probably, this is because they are closely related to other predictors with stronger effects, i.e., macroclimate and ecoregions, respectively[3]. The increasing influence of macroclimate and ecoregional species pool with grain size is, however, in line with evidence on the role of climatic and geological histories of ecoregions on species pools [8,10,20,24]. “

Fifthly, I don't know if the authors could perform some more sensitivity analyses for their results to explore other spatial scales and species components, but it seems that some of their results could still be artefacts of the analysis. For me, the two main findings from this research are that: fine local-scale α -diversity can be much more similar between tropical and temperate regions than differences in regional γ -diversity would suggest; but that while some areas showed consistent patterns of local species richness across grains, other areas did not. Therefore, to what extent might it be the case that areas differing in local diversity patterns across scales are caused by the particular grain sizes used or the missing species components in the data? For example, perhaps not having major species components such as epiphytes or herb layers in forest plots could explain consistent scale-independent diversity patterns, or conversely inconsistent scale-dependent diversity patterns, or might analysing species richness at still larger scales show a different pattern to those at smaller scales in either case?

Response: While we cannot push our models to larger spatial grains, since very few plots in our database counted the number of species in plots bigger than 1ha, we agree that our results might change substantially if one decided to include or exclude specific species groups. For instance, excluding vascular epiphytes clearly penalizes the tropics, as epiphytes are very rare (or absent) in the temperate and boreal regions. Alternatively, ecologists from the boreal region might be tempted to also include mosses and lichens, which represent a substantial share of the total diversity in cold regions. Ultimately, our choice was dictated by data availability, as we needed a consistent set of species across all climatic zones. In the revised manuscript, we discuss the problem related to the choice of the target functional groups, by better comparing our results to those obtained from using specific species groups, such as tree species, as in Keil & Chase (2019), habitats (e.g., alpine vegetation from Testolin et al., 2020, palearctic grasslands in Biurrun et al. 2021). Please see L381: “There is substantial agreement between our map of 1-ha local species richness in forests and a recently published global map of tree species richness at the same grain[9]. Similarly, patterns of fine-scale local species richness in non-forest ecosystems are consistent with the local and regional patterns recently observed for alpine vegetation[38] and Palearctic grasslands[25]”

We also created a new graph to show the performance of our models in case of incomplete sampling as suggested by referee 2 (see Supplementary Figure 5C). The comparison of the

frequency distribution of observed and predicted local species richness shows that our model is quite robust to incomplete sampling. The only exception is for small plots (<150m²) where only trees were sampled. This is not surprising, because such small plots are extremely rare in our database and are clearly inadequate to effectively sample tree biodiversity. In a sense, this mismatch supports our choice of only focusing on larger grains (>400 m²) for forests.

There are clearly very strong relationships between local α -diversity and regional γ -diversity patterns, but I was disappointed that the Discussion does not go into any real detail on these (despite citing several studies of regional γ -diversity patterns in the Introduction). The obvious missing piece of the picture, though, that is barely addressed in this paper, is clearly β -diversity, and how this varies spatially. Beyond citing a single study, (Condit, R. et al. Beta-Diversity in Tropical Forest Trees. Science 295, 666-669) as an explanation for diversity patterns in a single region (western Amazonia), β -diversity is hardly mentioned in the Discussion at all. As spatial scales increase and local species richness tends towards regional species richness, clearly this is due to the effect of β -diversity bringing more new species into the orbit of local species estimates. So the issue for me is less about 'how does α -diversity vary spatially?' but more about 'why does β -diversity vary so much between the tropics and the temperate regions?', and 'how and why does it also vary within both temperate and tropical regions?', because if local α -diversity is mostly the product of plot size and regional species pool, then β -diversity is what is causing the differences between local α -diversity and regional γ -diversity. I realise that the data the authors use for this study does not lend itself to analyses of β -diversity, but even if they cannot analyse this issue themselves it seems obtuse to barely even refer to this phenomenon in the Discussion. Word limits are always an issue in papers like these, but this seems such a glaring oversight that some less-essential text could surely be sacrificed for better discussion of the role of β -diversity: it is almost as if the wrong question has been asked. Partly this is itself a reflection of the paper not really being completely explicit that it only attempts to present and explain patterns of plant α -diversity (as opposed to 'local' diversity), which therefore makes the omission of β -diversity seem all the more glaring.

Response: We now state explicitly that our analysis focuses on local species richness (i.e. alpha diversity), which from the referee's comments should help address the problem. As the referee notes, our data are not well suited to analyse beta diversity, and we consider that attempting this would draw considerable criticism, as well as complicating the paper. Instead, we refer to beta diversity in our manuscript to provide explanations for the regional scaling anomalies encountered, in line with the referee's comments. We do however recognize the importance of the referee's comments, here, and we now discuss the issues more in-depth than previously – for example, in the first paragraph of the discussion (L369). And we tackle the issue later on, in L423: "How local species richness scales with increasing grain size, however, depends on the dominant eco-evolutionary processes structuring local plant communities. In the case of western Amazonia, for instance, much of the regional (gamma) diversity depends on species turnover (beta-diversity), rather than on the coexistence of a high number of species at the same site[50]. This points to more uniform environmental conditions at fine scales and dispersal limitation at larger scales. Such conditions are also known from regions that have been geologically and climatically stable for a long time, such as Western Australia[51]. In contrast, comparably high plant species richness at fine grains but low plant species richness at coarse grains might be an indicator of

effective niche partitioning at fine scales and more homogeneous landscapes without dispersal barriers at coarse scales[52]. In addition, high species richness at fine scales also depends on plant size, as more small plants can coexist in a given grain size. Such conditions mainly occur in grasslands, e.g. in Eastern Australia.”

In summary therefore, I feel the paper would benefit very much from: stating exactly how many records and species were used in the analysis; being clear that by ‘local’ diversity the authors are confining themselves to analysing α -diversity patterns; stating clearly how the maps were derived, the spatial thresholds defined and presenting combined forest and non-forest results together somewhere; elaborating on the probable impact of missing species components, and the relationship between α -diversity and β -diversity, on the outcome of their analyses; and adding some additional sensitivity analyses and greater discussion of the relationship between α -diversity and γ -diversity patterns. If these issues could be addressed I would be very happy to see this paper published in Nature Communications.

Response: Thanks for the many useful insights. These helped us enormously to improve our manuscript.

REVIEWER COMMENTS

Reviewer #1 (Remarks to the Author):

This is my second review of the paper by Sabatini et al. and I really appreciate the authors' efforts to improve the manuscript since the first submission. I believe that the model is now much more robust than in the previous version, and I also appreciate the clarification with respect to Meyr's biodiversity hotspots. I however still have two major concerns with this paper. The first one, also acknowledged by the other reviewers, is the undersampling of tropical regions. In my previous review I pointed the Western Ghats of India as an example, and I acknowledge the authors' efforts to gather more data for this region. I however think that the same problem remains with the other tropical regions, especially for Africa and Asia. It is a good practice to mention data-poor regions in maps as the authors did in their revision, but it doesn't prevent the model against sampling bias between e.g. tropical and temperate regions. On another hand, I'm pretty sure that more data sets exist in tropical regions, and it's really a pity that only very few co-authors from central Africa and South-East Asia have been enrolled in this synthesis work. My second concern, already pointed in my previous review, is that there is almost no eco-evolutionary interpretation of the detected patterns. Of course I acknowledge the impressive amount of work to gather and clean such a database and to fit a robust model at global scale. But it remains that this paper is more in my view a data paper than an analysis paper. It is of course to the editor to decide whether such a contribution deserves publication in Nat. Comms.

Reviewer #2 (Remarks to the Author):

I congratulate the authors on their outstanding effort to solve issues pointed out in the review and provide well-explained responses to the comments.

This revised version of the manuscript is much improved. The additional data for tropical regions (mitigating the poor sampling), the more explicit recognition of the under-sampling of tropical regions and the uncertainty in the inferences, along with the additional bias-correction, were great improvements.

My remaining concern is the seeming overambition to make predictions for the entire world, despite the poor sampling in many parts of the world and the bias towards certain biomes. I would favour a prediction coverage limited to regions with reasonably good sampling (or excluding regions clearly under-sampled), so comparisons between regions would be more robust and less dependent on bias-correction algorithms. However, I recognize that the ideal compromise between coverage and robustness of the evidence is somewhat subjective and may prevent exploratory studies.

The statistical/resampling procedures applied to correct the biases and gaps in the data were outstanding. The study includes an honest account of the biases and uncertainty in the findings (e.g., lines 455-479). Thus, the readers seem fairly warned on critical points of uncertainty and can make their judgement.

Some conclusions seem robust even considering the poorly sampled regions. For example, the less pronounced difference in species richness between temperate and tropical regions at finer grains and more apparent at coarser grains seem robust (and it makes sense). Thus, I do not expect that more data would change that.

The summary statistics for observed data Table S3 and Figure S5 (contrasting the frequency distribution of observed vs predicted values) were good additions. It makes the work more transparent to the readers, allowing them to understand the data better and perhaps compare it with (future) similar studies.

Toning down the recommendations for conservation prioritization, though a simple fix,

was a critical improvement!

Finally, this study showcases a great dataset and brings new evidence to support ecological knowledge (e.g., lines 480-492). I congratulate the authors on their commitment to address the comments and improve the study to their best ability. Given all the above, I can recommend this work for publication.

Issues to consider:

Are there too many zero or near-zero among the observed numbers of species in Figure S5? It is unclear to me how so many plot surveys find almost no vascular plants even in species-rich ecosystems. What could explain this amount of low observed values?

Is it surprising that western parts of the USA show consistently low plant species richness?

**Tarciso Leão
Research Fellow at the Kew Gardens**

Reviewer #3 (Remarks to the Author):

The authors have made a number of useful improvements to their previously-submitted manuscript and this interesting paper is now substantially better. However, while the changes to the Methods in particular are welcome, I do not feel that the improvements made sufficiently address the concerns that have been made, particularly to the text of the Discussion. Most of the responses are rebuttals to sensible suggestions, setting out why the authors are not going to follow the reviewers' suggestions. Despite valiant efforts to increase the scope of data, the dataset is still terribly biased towards temperate regions and away from more diverse tropical regions, to the extent that this can only be considered a preliminary 'global' analysis.

The response on line 543 to the point about the analysis actually being carried out on a relatively small proportion of species, the size of the overall dataset being analysed notwithstanding, completely misses the point. It is important that the paper is explicit about the number of species actually analysed, not all those included in the whole database but later omitted from the analysis, but as stated previously this is a small proportion of the total number of vascular plant species (just over 50,000, or <15% of 350,000 vascular plant species known worldwide) - the proportion of 87% of species from the sPlot database being included in the analyses is irrelevant and should be removed.

Also, I find the justification of sticking to 'local species diversity' to be very unconvincing: I would be amazed if anyone taking the time to read this paper in Nature Communications was unaware of the well established hierarchy of 'alpha/beta/gamma' diversity, but feel that continued use of 'local' rather than 'alpha' diversity conflates both alpha and beta diversity at very small scales. Although it is true that using more than one spatial grain within the same analysis means that this is not exclusively 'alpha' diversity, no citation is given for the contention that 'local' is more commonly used and more understandable than 'alpha' diversity on Line 166. Indeed, this is rather the very point the discussion ought to focus on: differences between alpha and gamma diversity across the world, their main results, are driven more by differences in beta diversity between tropical and temperate regions than by genuine differences in alpha diversity: how does this vary and why should this be so? Even in the revised manuscript this foundational concept is skated over very briefly, whereas I think a much greater focus of the Discussion should be given over to this.

If the authors were to make a better attempt at incorporating rather than just waving

away these and other comments from the 3 reviewers then I would be willing to see this paper accepted for publication.

REVIEWER COMMENTS

Reviewer #1 (Remarks to the Author):

This is my second review of the paper by Sabatini et al. and I really appreciate the authors' efforts to improve the manuscript since the first submission. I believe that the model is now much more robust than in the previous version, and I also appreciate the clarification with respect to Meyr's biodiversity hotspots. I however still have two major concerns with this paper.

The first one, also acknowledged by the other reviewers, is the undersampling of tropical regions. In my previous review I pointed the Western Ghats of India as an example, and I acknowledge the authors' efforts to gather more data for this region. I however think that the same problem remains with the other tropical regions, especially for Africa and Asia. It is a good practice to mention data-poor regions in maps as the authors did in their revision, but it doesn't prevent the model against sampling bias between e.g. tropical and temperate regions. On another hand, I'm pretty sure that more data sets exist in tropical regions, and it's really a pity that only very few co-authors from central Africa and South-East Asia have been enrolled in this synthesis work.

Response: We are glad referee #1 appreciates our efforts. We also concede that tropical areas are still undersampled compared to temperate regions. Yet, this is a structural problem that can hardly be solved within the revision stage of any scientific paper. Over more than 8 years, the sPlot consortium poured uncountable hours trying to locate and integrate datasets from these undersampled areas. The dedicated additional data search we did during the previous round of revision clearly could only improve things marginally, but we are happy our effort is appreciated. Yet, please note that we did not only focus on Western Ghats (indeed, only ~200 out of the ~10,000 additional plots were from that region), but on all tropical areas. If we included only few co-authors from central Africa and SE Asia, however, is not for lack of trying, but for the existence of real barriers (language, difficulty of networking, lack of contacts) which hindered us to reach out to vegetation scientists from these regions over the years. We even used unconventional channels (i.e., social media) to recruit additional researchers and find additional data from these regions. In some cases, it worked, but we got little feedback from areas such as Subsaharian Africa and SE Asia. Yet, even if we are positive that more data exists from these regions, accessing these data remains challenging, and it is unlikely we will ever get a data coverage as dense as the one in Europe or N America any time soon. Expanding our network and the data availability is (un)fortunately a never-ending quest.

On one point, however, we disagree. We put all possible safeguards in place (e.g., stratified resampling, rarefaction of plots in oversampled areas, multiple runs, spatial-block validation) to train and assess our statistical models in a way that is as little impacted by the bias in data distribution as possible. Clearly, any strategy can only go this far at facing the problem of data scarcity, but in referee #2's words, "Some conclusions seem robust even considering the poorly sampled regions" and it is unlikely that "more data would change that".

My second concern, already pointed in my previous review, is that there is almost no eco-evolutionary interpretation of the detected patterns. Of course I acknowledge the impressive amount of work to gather and clean such a database and to fit a robust model at global scale. But it remains that this paper is more in my view a data paper than an analysis paper. It is of course to the editor to decide whether such a contribution deserves publication in Nat. Comms.

Response: We understand Referee #1's disappointment here, even if we perceive the critique of ours being a 'data paper, more than an analysis paper' as unjust. We trained a model to explore the underlying macroecological drivers explaining an aspect of biodiversity (alpha-diversity) that has never been explored at a global extent. It is, maybe, just an exploratory analysis, but an analysis that we believe will trigger intense debate and, hopefully, will become the foundation for new research on several aspects we could not deal with in our work. To partially add insights on the possible eco-evolutionary mechanisms underlying the diversity patterns we found, however, we followed referee #3's suggestion and included a new paragraph to the discussion (see below). We concede we are just scratching the surface here, but we hope this change goes in the direction suggested by referee #1.

Reviewer #2 (Remarks to the Author):

I congratulate the authors on their outstanding effort to solve issues pointed out in the review and provide well-explained responses to the comments.

This revised version of the manuscript is much improved. The additional data for tropical regions (mitigating the poor sampling), the more explicit recognition of the under-sampling of tropical regions and the uncertainty in the inferences, along with the additional bias-correction, were great improvements.

Response: Thanks for your help and suggestions on how to improve the first version of our manuscript!

My remaining concern is the seeming overambition to make predictions for the entire world, despite the poor sampling in many parts of the world and the bias towards certain biomes. I would favour a prediction coverage limited to regions with reasonably good sampling (or excluding regions clearly under-sampled), so comparisons between regions would be more robust and less dependent on bias-correction algorithms. However, I recognize that the ideal compromise between coverage and robustness of the evidence is somewhat subjective and may prevent exploratory studies.

The statistical/resampling procedures applied to correct the biases and gaps in the data were outstanding. The study includes an honest account of the biases and uncertainty in the findings (e.g., lines 455-479). Thus, the readers seem fairly warned on critical points of uncertainty and can make their judgement.

Response: After asking for the editor's advice here, we decided to only show in the main text the multi-scale maps where the undersampled regions are excluded. Equivalent maps also showing these undersampled regions were moved to the Supplementary Material. Many thanks for appreciating our approach, and understanding the difficulty of working with such large databases.

Some conclusions seem robust even considering the poorly sampled regions. For example, the less pronounced difference in species richness between temperate and tropical regions at finer grains and more apparent at coarser grains seem robust (and it makes sense). Thus, I do not expect that more data would change that.

The summary statistics for observed data Table S3 and Figure S5 (contrasting the frequency distribution of observed vs predicted values) were good additions. It makes the work more transparent to the readers, allowing them to understand the data better and perhaps compare it with (future) similar studies.

Toning down the recommendations for conservation prioritization, though a simple fix, was a critical improvement!

Finally, this study showcases a great dataset and brings new evidence to support ecological knowledge (e.g., lines 480-492). I congratulate the authors on their commitment to address the comments and improve the study to their best ability. Given all the above, I can recommend this work for publication.

Response: Thanks for the many insightful suggestions, and these encouraging words.

Issues to consider:

Are there too many zero or near-zero among the observed numbers of species in Figure S5? It is unclear to me how so many plot surveys find almost no vascular plants even in species-rich ecosystems. What could explain this amount of low observed values?

Response: This is a fair point. We took a close look at the data and tried to figure out where these low numbers come from. Our original dataset contained ~2400 plots (1.4% of the total) with species richness < 3. These plots seem to be distributed across a wide range of regions and habitats. The image below shows their spatial distribution.

A systematic classification of these plots into habitats is, unfortunately, not available. But after taking a look at the species contained in these plots, we realized two things:

- 1) Our dataset still contained some (species-poor) plots from aquatic environments. We, therefore, refined our filters by creating a list of aquatic species, and excluded all those plots where these species accounted for more than 50% of relative cover ($n=4720$; not all have species richness < 3).
- 2) The remaining plots seem to belong to naturally species-poor habitats, such as: 1) coastal cliff and dune habitats, 2) temporary inundated grasslands, 3) semi-desertic rangeland (e.g.,

those in the Western US, Morocco, Australia), 4) heathland, 5) species-poor forests on acidic soils dominated by *Fagus sylvatica* or *Pinus sylvestris*. We kept these plots in the model.

We updated all graphs and tables based on the new subset of plots. Results are hardly any different from those shown in the first revision of the paper.

Is it surprising that western parts of the USA show consistently low plant species richness?

Response: Again, we don't have a good answer to this question. We looked into the data to figure out whether this depends on some sort of artefacts. For the region of NW USA (40-50° N, 103-125°W) we have ~8200 plots, 200 of which have species richness < 3. This is 2.1% of plots, very close to the global level (1.4%), although we found very few aquatic plots. Almost all of these plots are of average size (400 m²). After a careful examination of the plots, we found 113 invalid vegetation plots, i.e., plots with clearly incomplete species lists, which we excluded before rerunning all the analyses.

When looking at the species composing the remaining plots, the most common species are either associated with habitats at the land-water interface (e.g., *Carex utriculata*, *Equisetum fluviatile*), forested swamps (*Thuja plicata*), fire-dominated dry mountain forests (*Pinus contorta*, *Larix occidentalis*), disturbed areas (*Xerophyllum tenax*), as well as exotic species (*Poa pratensis*). These are all ecosystems where a species richness <3, while uncommon, is plausible, in our view. We decided, therefore, to keep these plots in the models, as we believe that low predicted species richness in this region does not depend on an overrepresentation of species-poor habitats in the data.

Tarciso Leão
Research Fellow at the Kew Gardens

Reviewer #3 (Remarks to the Author):

The authors have made a number of useful improvements to their previously-submitted manuscript and this interesting paper is now substantially better. However, while the changes to the Methods in particular are welcome, I do not feel that the improvements made sufficiently address the concerns that have been made, particularly to the text of the Discussion. Most of the responses are rebuttals to sensible suggestions, setting out why the authors are not going to follow the reviewers' suggestions. Despite valiant efforts to increase the scope of data, the dataset is still terribly biased towards temperate regions and away from more diverse tropical regions, to the extent that this can only be considered a preliminary 'global' analysis.

Response: We are glad referee #3 appreciates the improvements of our previously-submitted manuscript, and we are thankful for the many useful suggestions.

In this version we tried to address the remaining issues related to the discussion, by 1) openly disclosing that our dataset includes ~15% of the estimated number of plant species, 2) changing the terminology as suggested by referee #3, from local-species richness to alpha diversity, 3) giving more prominence to the issue of beta diversity in the discussion. Please see below for additional details.

The response on line 543 to the point about the analysis actually being carried out on a relatively

small proportion of species, the size of the overall dataset being analysed notwithstanding, completely misses the point. It is important that the paper is explicit about the number of species actually analysed, not all those included in the whole database but later omitted from the analysis, but as stated previously this is a small proportion of the total number of vascular plant species (just over 50,000, or <15% of 350,000 vascular plant species known worldwide) - the proportion of 87% of species from the sPlot database being included in the analyses is irrelevant and should be removed.

Response: This is a fair point. We corrected it as: "Altogether, these plots provided 9,953,940 occurrence records for 53,271 vascular plant taxa, i.e., ~15% of ~350,000 plant species estimated to exist. This figure is slightly underestimated, since for 1,893 plots (59,299 occurrence records) only aggregated alpha diversity data were available, but no species-level data."

Also, I find the justification of sticking to 'local species diversity' to be very unconvincing: I would be amazed if anyone taking the time to read this paper in Nature Communications was unaware of the well established hierarchy of 'alpha/beta/gamma' diversity, but feel that continued use of 'local' rather than 'alpha' diversity conflates both alpha and beta diversity at very small scales. Although it is true that using more than one spatial grain within the same analysis means that this is not exclusively 'alpha' diversity, no citation is given for the contention that 'local' is more commonly used and more understandable than 'alpha' diversity on Line 166. Indeed, this is rather the very point the discussion ought to focus on: differences between alpha and gamma diversity across the world, their main results, are driven more by differences in beta diversity between tropical and temperate regions than by genuine differences in alpha diversity: how does this vary and why should this be so? Even in the revised manuscript this foundational concept is skated over very briefly, whereas I think a much greater focus of the Discussion should be given over to this.

Response: We concede on this point. We modified our wording and switched to the use of the alpha/beta/gamma terminology. We hope the manuscript is now clearer. Also, the discussion now includes an extensive explanation of the possible factors causing the differences in beta-diversity we observed between, e.g., lowland tropics vs temperate forests. The debate on this topic is quite intense in the literature at the moment and we caught the occasion to link our results to the findings deriving from many recently published, exciting papers. We thank the referee for stimulating us on creating these new connections. Please see the fourth and fifth paragraphs of the discussion:

"In general, finding these scaling anomalies points to the role of beta diversity as a cross-scale diversity metrics, and suggests that the relative contribution of different eco-evolutionary processes in determining plant diversity patterns varies between regions. In many tropical lowland forests, alpha diversity is low at fine grains but increases rapidly with increasing grain size. This is the case, for instance, in the western Amazon, where much of the regional (gamma) diversity depends on species turnover rather than on the coexistence of a high number of species at the same site[52]. This suggests that the tropics might be shaped by processes promoting species coexistence through a tighter packing in the niche space. Recent work found a latitudinal increase in niche specialization and marginality of trees towards the equators, which has been attributed to the stable climate and high productivity in the tropics[53]. Alternative explanations include rarity and priority effects related to high productivity[29], more uniform environmental conditions and stronger dispersal limitation at fine scales[28], or stronger mycorrhiza-mediated effects of interspecific competition and habitat adaptation[54] in the tropics compared to temperate regions.

Conversely, we observed high plant alpha diversity at fine grains but relatively low alpha diversity at coarse grains in many temperate regions, including the Eastern European forest-steppe belt, East Asia

and southeastern Australia. This pattern might be indicative of effective niche partitioning at fine grains and more homogeneous landscapes without dispersal barriers at coarse grains[55]. There is evidence that niche processes play a stronger role than neutral processes in determining fine-scale beta diversity at higher latitudes and altitudes[28-30], where species are thought to have broader niches and be less responsive to geographical changes[56]. This is consistent with recent findings that the nestedness of tree communities increases with latitude, possibly due to the high share of ectomycorrhizal species in colder and wetter conditions[54]. Finally, high species richness at fine grains might also depend on plant size, as many small plants can coexist in a given grain size. Such conditions mainly occur in grasslands, e.g., in Eastern Australia, where this mechanism has been invoked to explain differences in beta diversity among vegetation types[57]. “

If the authors were to make a better attempt at incorporating rather than just waving away these and other comments from the 3 reviewers then I would be willing to see this paper accepted for publication.

Response: Thanks for your time and the many useful insights

REVIEWERS' COMMENTS

Reviewer #1 (Remarks to the Author):

My opinion with this paper didn't change throughout revisions: it lacks ecological hypotheses and analysis to really be a great paper. I however acknowledge the authors' efforts to make the mapping model more robust or at least to have indicated areas (mostly tropical) where uncertainty is high and the maps should be interpreted with caution. I also acknowledge the attempt to introduce insights in the discussion that suggest explanatory directions for the observed patterns and I hope this first contribution will stimulate further questioning and research in a near future.

Reviewer #2 (Remarks to the Author):

I appreciate the changes made by the authors, e.g., the maps presented in Figure 1 look more reasonable, given my expectation from the theoretical knowledge. The authors maintained key improvements, including the honest account of limitations (lines 467-491). As pointed out in my last review, this study showcases a great new dataset, and it brings interesting perspectives and evidence, such as the scaling anomalies in alpha diversity and its consequences for understanding biodiversity patterns and implications for macroscale conservation planning. Future studies are likely to provide better estimates, particularly as data from the tropics become more available. Still, this study seems to make a fair contribution to improving our knowledge. Thus, I uphold my recommendation to publish this study.

Reviewer #3 (Remarks to the Author):

This manuscript is now much improved and I would be willing to see this accepted for publication barring a few final, minor issues set out below.

Although residual issues remain with regards to geographical biases in the scope of data and thus robustness of analyses, the authors have made a valiant attempt to address these as far as they are able to. While it is clearly an enormous amount of work, the study should therefore be seen as a 'preliminary' global analysis for these reasons, pending more representative data from tropical regions.

The new paragraphs on beta diversity added to the Discussion are welcome - for me, this is very clearly the main finding of the paper: alpha diversity of plant species may be similar at fine scales in temperate and in tropical regions, but gamma diversity is much greater in tropical regions than in temperate regions. The only possible explanation for this is that beta diversity is much greater in tropical regions than it is in temperate regions.

Why this should be is not yet understood, in my opinion. Personally I was unconvinced by any of the explanations for greater tropical beta diversity that were mentioned in the paper, although it was interesting to see these all set out. I can't see that any of the possible explanations for finer niche use in the tropics (and hence higher beta diversity) such as topographic diversity or increased mycorrhizal diversity are really so different between tropical and temperate ecosystems. Currently, this is all speculation and it would be nice to see this acknowledged as such.

I would rather see the authors state that a better understanding of the spatial variation of beta diversity should now be the objective of plant diversity analyses. I do not mean to downplay their own contribution to this with this study of alpha diversity in any way. They have convincingly demonstrated that differences between temperate and tropical diversity of plant species are due to the spatial variation in beta diversity rather than spatial differences in alpha diversity, and this is a great step forward. It's just that spatial differences in alpha diversity are not the driver of spatial differences in gamma diversity on which much ecological understanding and conservation is based: spatial differences in beta diversity are.

This is an additional reason why I much prefer to keep the established terminology of 'alpha',

'beta' and 'gamma' diversity rather than refer to 'local' as opposed to 'regional' plant diversity (and for which the authors should be thanked for conceding on this point): there is a reason why the term beta diversity was introduced - to disentangle different factors operating at 'local' scales - so it is an unnecessarily retrograde step to revert back to 'local' diversity, it just confuses two separate issues.

Please also note that on line 422 in the new material added to the Discussion there is a typo: it should be "a cross-scale diversity metric" not "metrics".

REVIEWERS' COMMENTS

Reviewer #1 (Remarks to the Author):

My opinion with this paper didn't change throughout revisions: it lacks ecological hypotheses and analysis to really be a great paper. I however acknowledge the authors' efforts to make the mapping model more robust or at least to have indicated areas (mostly tropical) where uncertainty is high and the maps should be interpreted with caution. I also acknowledge the attempt to introduce insights in the discussion that suggest explanatory directions for the observed patterns and I hope this first contribution will stimulate further questioning and research in a near future.

Thanks for reviewing our paper, and for the fair, although critical, points raised on our work. They did stimulate us to do better.

Reviewer #2 (Remarks to the Author):

I appreciate the changes made by the authors, e.g., the maps presented in Figure 1 look more reasonable, given my expectation from the theoretical knowledge. The authors maintained key improvements, including the honest account of limitations (lines 467-491). As pointed out in my last review, this study showcases a great new dataset, and it brings interesting perspectives and evidence, such as the scaling anomalies in alpha diversity and its consequences for understanding biodiversity patterns and implications for macroscale conservation planning. Future studies are likely to provide better estimates, particularly as data from the tropics become more available. Still, this study seems to make a fair contribution to improving our knowledge. Thus, I uphold my recommendation to publish this study.

Many thanks for the insightful review, and for the appreciation of our work.

Reviewer #3 (Remarks to the Author):

This manuscript is now much improved and I would be willing to see this accepted for publication barring a few final, minor issues set out below.

Although residual issues remain with regards to geographical biases in the scope of data and thus robustness of analyses, the authors have made a valiant attempt to address these as far as they are able to. While it is clearly an enormous amount of work, the study should therefore be seen as a 'preliminary' global analysis for these reasons, pending more representative data from tropical regions.

The new paragraphs on beta diversity added to the Discussion are welcome - for me, this is very clearly the main finding of the paper: alpha diversity of plant species may be similar at fine scales in temperate and in tropical regions, but gamma diversity is much greater in tropical regions than in temperate regions. The only possible explanation for this is that beta diversity is much greater in tropical regions than it is in temperate regions.

We are glad referee 3 appreciates the new paragraph, and agree with the suggested interpretation of the matter.

Why this should be is not yet understood, in my opinion. Personally I was unconvinced by

any of the explanations for greater tropical beta diversity that were mentioned in the paper, although it was interesting to see these all set out. I can't see that any of the possible explanations for finer niche use in the tropics (and hence higher beta diversity) such as topographic diversity or increased mycorrhizal diversity are really so different between tropical and temperate ecosystems. Currently, this is all speculation and it would be nice to see this acknowledged as such.

This is a fair point. We added a sentence to indicate that the debate on the main processes behind the high species turnover in the tropics remains unresolved.

I would rather see the authors state that a better understanding of the spatial variation of beta diversity should now be the objective of plant diversity analyses. I do not mean to downplay their own contribution to this with this study of alpha diversity in any way. They have convincingly demonstrated that differences between temperate and tropical diversity of plant species are due to the spatial variation in beta diversity rather than spatial differences in alpha diversity, and this is a great step forward. It's just that spatial differences in alpha diversity are not the driver of spatial differences in gamma diversity on which much ecological understanding and conservation is based: spatial differences in beta diversity are.

We agree and added a sentence to the discussion to stress this important implication of our work:

“While the relative contribution of these processes remains a matter of speculation, our work points to the need for an improved understanding of the spatial variation of beta diversity in plant diversity analysis⁵³. Beta diversity, rather than alpha diversity per se, appears to be the main driver of spatial differences in gamma diversity between temperate and tropical regions.”

This is an additional reason why I much prefer to keep the established terminology of 'alpha', 'beta' and 'gamma' diversity rather than refer to 'local' as opposed to 'regional' plant diversity (and for which the authors should be thanked for conceding on this point): there is a reason why the term beta diversity was introduced - to disentangle different factors operating at 'local' scales - so it is an unnecessarily retrograde step to revert back to 'local' diversity, it just confuses two separate issues.

Again a fair point. We further reinforced the terminological consistency of our ms by correcting a couple of passages in the discussion where we were still referring to 'local' or 'regional' species richness rather than alpha\gamma diversity by mistake.

Please also note that on line 422 in the new material added to the Discussion there is a typo: it should be "a cross-scale diversity metric" not "metrics".

Corrected thanks.Thanks for that and thanks for your time and stimulating comments.